# Bicontinuous RuO$_2$ nanoreactors for acidic water oxidation

Ding Chen[1,4], Ruohan Yu[1,2,4], Kesong Yu [1,4], Ruihu Lu[1,4], Hongyu Zhao[1], Jixiang Jiao[1], Youtao Yao[1], Jiawei Zhu[1], Jinsong Wu [1,3] & Shichun Mu [1] ✉

Improving activity and stability of Ruthenium (Ru)-based catalysts in acidic environments is eager to replace more expensive Iridium (Ir)-based materials as practical anode catalyst for proton-exchange membrane water electrolyzers (PEMWEs). Here, a bicontinuous nanoreactor composed of multiscale defective RuO$_2$ nanomonomers (MD-RuO$_2$-BN) is conceived and confirmed by three-dimensional tomograph reconstruction technology. The unique bicontinuous nanoreactor structure provides abundant active sites and rapid mass transfer capability through a cavity confinement effect. Besides, existing vacancies and grain boundaries endow MD-RuO$_2$-BN with generous low-coordination Ru atoms and weakened Ru-O interaction, inhibiting the oxidation of lattice oxygen and dissolution of high-valence Ru. Consequently, in acidic media, the electron- and micro-structure synchronously optimized MD-RuO$_2$-BN achieves hyper water oxidation activity (196 mV @ 10 mA cm$^{-2}$) and an ultralow degradation rate of 1.2 mV h$^{-1}$. A homemade PEMWE using MD-RuO$_2$-BN as anode also conveys high water splitting performance (1.64 V @ 1 A cm$^{-2}$). Theoretical calculations and in-situ Raman spectra further unveil the electronic structure of MD-RuO$_2$-BN and the mechanism of water oxidation processes, rationalizing the enhanced performance by the synergistic effect of multiscale defects and protected active Ru sites.

Under the blueprint of the hydrogen economy, using intermittent renewable energy to drive proton-exchange membrane water electrolyzers (PEMWEs) is attractive for green hydrogen production due to high efficiency and safety[1–4]. However, to realize the large-scale commercial application of PEMWEs, the development of high-performance and low-cost catalysts becomes a critical bottleneck[5–7]. This is because the oxygen evolution reaction (OER) at anode involves a sluggish four-electron transferred kinetics and thermodynamic uphill process, requiring much high energy barriers to facilitate the reaction[8–11]. Furthermore, except for very expensive iridium (Ir)-based catalysts, most existing OER catalysts are difficult to work continuously and stably under harsh acidic and oxidative environments in PEMWEs[12–14]. Therefore, we have to explore high cost-performance acid OER catalysts.

By comprehensive consideration of price, abundance and activity, the rutile-structured ruthenium oxide (RuO$_2$)-based catalysts would the best choice for anode catalysts of PEMWEs[15,16]. However, the biggest obstacle is the unsatisfactory long-term stability[17]. Because, the oxidation process of the catalytic site would cause the formation and dissolution of high-valence Ru species (RuO$_4$$^{2-}$ and RuO$_4$) and the participation of lattice oxygen. Both the active site loss and crystal structure change lead to the rapid inactivation of RuO$_2$[18–20]. To date, tremendous strategies, such as morphology and structure tuning[16,21], lattice doping[20,22,23], alloying[24–26] and even defect engineering[27–29], have been devoted to improving the performance of RuO$_2$. Regretfully, there is very little work done to concern the RuO$_2$ nano-micro reactors with more efficient active sites, which not only inhibits the inactivation

[1]State Key Laboratory of Advanced Technology for Materials Synthesis and Processing, Wuhan University of Technology, Wuhan 430070, China. [2]The Sanya Science and Education Innovation Park of Wuhan University of Technology, Sanya 572000, China. [3]NRC (Nanostructure Research Centre), Wuhan University of Technology, Wuhan 430070, China. [4]These authors contributed equally: Ding Chen, Ruohan Yu, Kesong Yu, Ruihu Lu. ✉e-mail: msc@whut.edu.cn

of Ru species but also boost the intrinsic activity of RuO$_2$[30,31]. On the other hand, the construction of a suitable reactor is conducive to the enhancement of the atomic utilization and economy, as well as the mass transfer and gas release, so as maintaining the efficient and stable operation of catalysts[32–35]. Therefore, such an integrated design of electronic structures and microstructures is highly desirable for practical applications of PEMWEs, yet challenging in design, synthesis and analysis in the fine structure-activity relationship.

Herein, we pioneer the use of the liquid molten salt to trigger the dual modulation of Ru electronic characteristics and local microenvironments, and construct a nanoreactor with RuO$_2$ nanocrystals as matrix. By means of the three-dimensional (3D) tomograph reconstruction technology, it shows a bicontinuous structure for the nanoreactor composed of ultrafine RuO$_2$ nanomonomers, which not only provides abundant active sites and reaction regions for catalysis, but also enhances mass and electron transfer through a cavity confinement effect. Furthermore, the in-depth characterization analysis indicates that there are multiscale defects such as vacancies and grain boundaries in RuO$_2$ particles, which change the local electronic structure and coordination environment of Ru, and then weakens the Ru-O interaction. The resulting bicontinuous nanoreactor consisting of multiscale defective RuO$_2$ nanomonomers (MD-RuO$_2$-BN), with simultaneous optimization of electron- and micro-structures, indeed exhibits good acidic OER performance in electrochemical tests. Also, we prove a high-performance PEMWE with MD-RuO$_2$-BN at anode, which outputs a very low cell voltage of 1.64 V at a the current density of 1 A cm$^{-2}$. Finally, the density functional theory (DFT) calculations confirm the synergistic effect of the multiscale defects on enhancing RuO$_2$ activity and stability. This work demonstrates a feasible idea to design and synthesize a highly-efficient and stable Ru-based catalyst with rich nanoreactors, offering a great promise for practical applications in PEMWEs.

## Results

### Structural design and characterization

Figure 1a shows the scheme of the fabrication and catalytic procedure for MD-RuO$_2$-BN. As the liquid trigger of reactions in Ar atmosphere at 500 °C, the KCl-LiCl molten salt system[36,37], with the lowest eutectic point of 352 °C (Fig. S1), induces the formation of RuO$_2$ nanoparticles with multiscale defects (including Ru and O vacancies, and inter and intra granular boundaries) through corrosion. Meanwhile, thanks to the fluidity and uniformity of the molten salt at high temperatures[38–40], when the reaction is cooled and the recrystallized salt is removed, the ultrafine RuO$_2$ grain and internal pore would be assembled into bicontinuous nanoreactors, thus driving water oxidation efficiently and stably in acidic media. Moreover, this facile synthesis route not only yields up to 91% but also is easily expanded to a relatively large scale production (Fig. S2), crucial to promote the commercial production of MD-RuO$_2$-BN catalysts.

Figure 1b, c shows the microstructure of the catalyst by double spherical aberration-corrected scanning transmission electron microscope (AC-STEM). By local magnification, we can find that MD-RuO$_2$-BN is composed of generous interconnected nanomonomers with the average size of only about 3 nm (Figs. 1c and S3, S4). Here, ultrafine grains and rich inter granular boundaries would allow MD-RuO$_2$-BN to have higher active site leakage ratios and faster interfacial charge transfer[29,41]. In addition, the coexistence of Ru and O atoms, and their homogeneous distribution over entire catalyst are disclosed by electron energy loss spectroscopy (EELS) of a single crystal (inset of Fig. 1c) and energy dispersive spectrum (EDS) elemental mapping (Figs. 1d and S5). The X-ray diffraction (XRD) pattern (Fig. 1e) and corresponding crystal structure (Fig. S6), and lattice spacing (0.25 and 0.32 nm) displayed by a high-resolution transmission electron microscope (HRTEM) image (Fig. S7), further confirm the acquisition of the rutile-structured RuO$_2$. Notably, the diffraction peak of MD-RuO$_2$-BN becomes wider and weaker than that of commercial RuO$_2$ (C-RuO$_2$),

implying the formation of small-sized nanoparticles[42,43], consistent with the above TEM observation results. For comparison, Fig. S8 exhibits that C-RuO$_2$ possesses larger and uneven grain size, and obviously disorganized and agglomerated spatial structures. The typical aggregation of homogeneous nanocrystals for MD-RuO$_2$-BN is also reflected in its typical type II isotherm and H1 type hysteresis loop (Fig. 1f)[22]. The corresponding N$_2$ adsorption-desorption measurements show the surface area of 69.6 m$^2$ g$^{-1}$ for MD-RuO$_2$-BN, much larger than that of C-RuO$_2$ (14.3 m$^2$ g$^{-1}$). And the Barrett-Joyner-Halenda pore diameter of MD-RuO$_2$-BN and C-RuO$_2$ is ~6.1 and 15.2 nm (inset of Fig. 1f), respectively. The above results imply construction of a suitable microstructure with rich and highly dispersed active sites toward catalysis.

To further gain insight into the internal microenvironment of MD-RuO$_2$-BN, 3D tomography reconstruction was carried out. As shown in Supplementary Movie 1, ~90 STEM-HAADF images for tomography reconstruction of representative position were collected by a 1–2° interval, over preferably 140°. The resulting reconstructed structural units and some HAADF images at different rotation angles are displayed in Figs. 2a and S9. The reconstructed shape at front view is roughly similar to HAADF image taken at middle angles (70°), proving the effectiveness and authenticity of this reconstruction. Figure 2b–d exhibits the front, top and right view images of the reconstruction unit, and the length (x axis), width (z axis) and height (y axis) of the bonding box are 80, 80 and 120 nm, respectively. Moreover, the contrast of the volume is presented in physics color mode where the volume in blue/green represents relatively high/low HAADF contrast. Since the catalyst only contains the RuO$_2$ component, the high and low contrast in the structure corresponds to RuO$_2$ nanoparticles and interspace, respectively. The representative ortho slice (Fig. 2e) marked by black dash line in the right view of the reconstruction unit (xy planes, perpendicular to the z axis at 18, 34, 50, 66 nm) further indicates the presence of unique bicontinuous nanoreactors composed of ultrafine RuO$_2$ nanomonomers and internal pores. In addition, the rotation, filtering and ortho-slicing dynamic process of the reconstructed particle are presented for a better view (Supplementary Movie 2).

Next, to further confirm the bicontinuous nanoreactor structure, an internal sub-volume cube with an edge length of about 30 nm (Figs. 2f and S10) was extracted from the reconstructed structural unit (yellow dash line marked area in Fig. 2b–d). The contrast was converted to VolrenRed for a better visualization of the inner structure (Supplementary Movie 3). Both the surface (Figs. 2g and S11) from different view directions, and the inner structure (Fig. 2h) from ortho slices are also identified to be bi-continuous. Furthermore, as shown in Fig. 2i, to separate the high contrast volume (RuO$_2$) and low contrast one (interspace), a segmentation is conducted by the STEM-HAADF contrast. Both volumes extracted from contrast segmentation are proven to be continuous (Figs. S12 and 13 and Supplementary Movie 4), providing direct evidence for construction of bicontinuous nanoreactors. The formation of such a special structure facilitates the penetration of electrolyte in the catalytic process, and then the broad active interface for the reaction can be realized. As a good ion-electron transport channel, the internal continuous pore would provide a localized environment through the regionalization constraint and enrichment effects, thereby strengthening mass transfer and optimizing the adsorption and conversion of active intermediates[44,45]. Meanwhile, the bicontinuous nanoreactor is expected to avoid the accumulation of catalysts and enhance the ability of gas release, ensuring the high OER catalytic activity and stability.

In addition, to demonstrate the role of eutectic solutions in creating the unique bicontinuous structured nanoreactor and further analyze the formation mechanism, we conducted a control experiment using KCl instead of the KCl-LiCl eutectic system under the same condition (K-RuO$_2$, Fig. S14). Due to the higher melting point of KCl (770 °C) than the synthesis temperature (500 °C), the reaction system

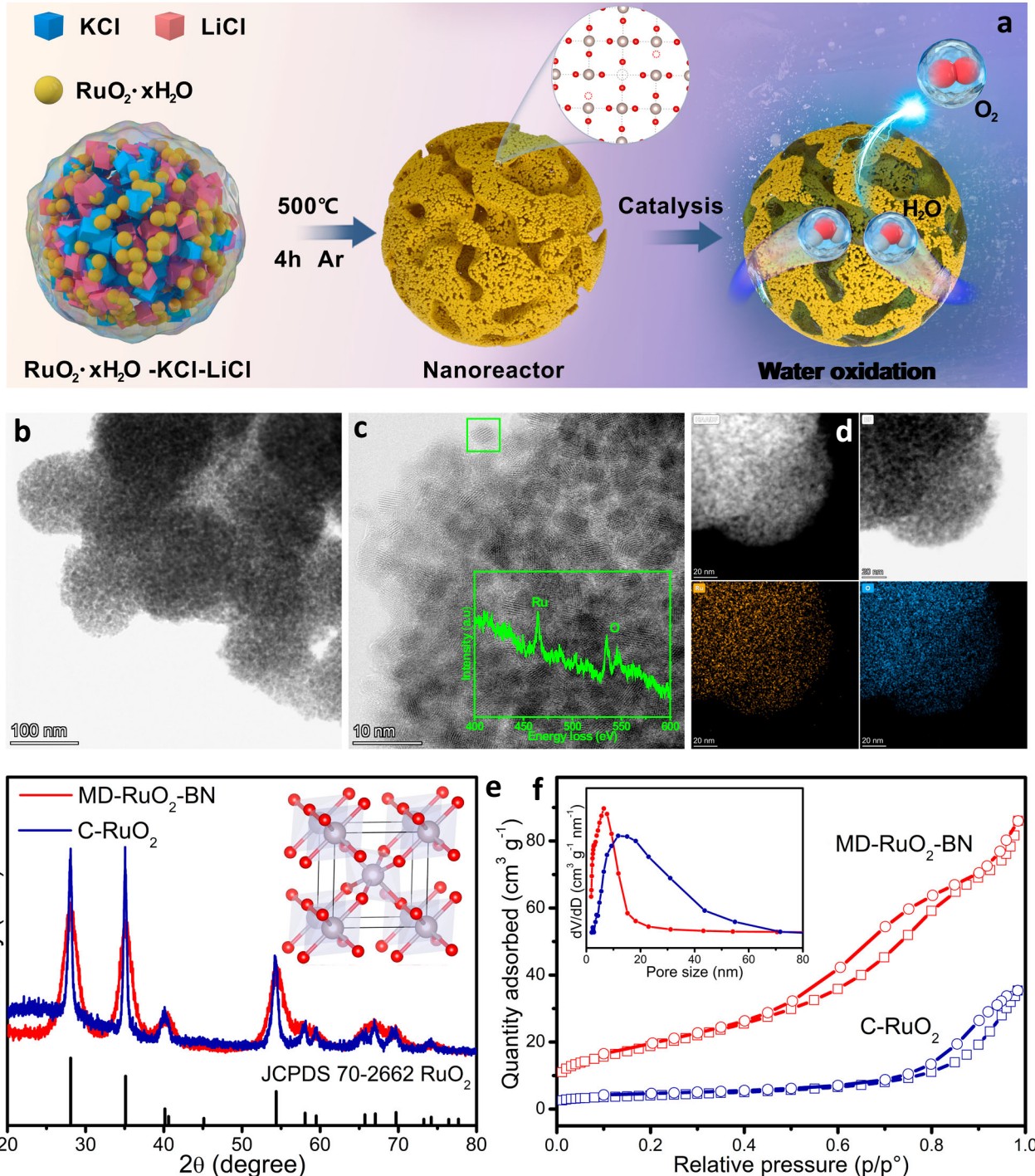

**Fig. 1 | Construction and structure characterization of catalysts. a** Schematic illustration of the fabrication and catalytic procedure of MD-RuO$_2$-BN. **b**, **c** STEM images of MD-RuO$_2$-BN. Inset: EELS analysis of a single crystal. **d** STEM mapping and corresponding elemental distribution of MD-RuO$_2$-BN. **e** XRD patterns of MD-RuO$_2$-BN and C-RuO$_2$. **f** N$_2$ adsorption-desorption isotherm and pore size distribution plot of MD-RuO$_2$-BN and C-RuO$_2$.

is unable to produce a liquid reaction medium. Corresponding nanostructure characterizations (Figs. S15 and 16) show that the average grain size of K-RuO$_2$ (~10 nm) is reduced relative to C-RuO$_2$ (~20 nm), but still much larger than that of MD-RuO$_2$-BN (~3 nm). Importantly, due to lack of a liquid medium that can provide a uniform growth environment, the grain size of K-RuO$_2$ is uneven and disorderly arranged. Moreover, the 3D tomography reconstruction result on K-RuO$_2$ (Figs. S17–19) also indicates that K-RuO$_2$ does not possess a nanoreactor structure like in MD-RuO$_2$-BN. Therefore, the liquid environment and structure-oriented effects generated by eutectic

molten salts induce the formation of ultrafine, uniform and continuous RuO$_2$ nanomonomers, and periodically regular porous structures with 3D interconnections (Fig. S20), which are important for formation of bicontinuous nanoreactors.

## Atomic and local electronic structures

We further conducted in-depth observation of MD-RuO$_2$-BN on the atomic scale. STEM images (Figs. 3a and S21) reveal that some lattice positions are blurred or even missing (marked by blue circle), indicating the existence of Ru vacancy defects (V$_{Ru}$), which is further

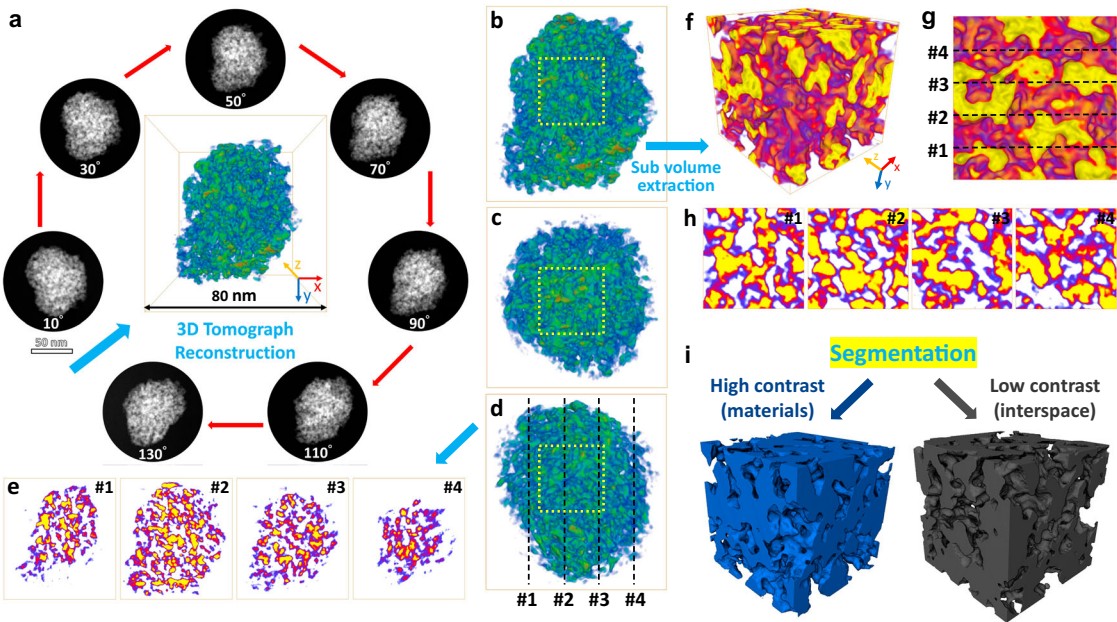

**Fig. 2 | Illustration of MD-RuO₂-BN by 3D tomography reconstruction.**
**a** Representative STEM-HAADF images at different rotation angles and reconstructed MD-RuO₂-BN at front view. **b–d** Corresponding front, top and right view of reconstructed MD-RuO₂-BN. **e** Representative ortho slices marked by black dash line in (**d**) (*xy* planes, perpendicular to the *z* axis at 18, 34, 50, 66 nm). **f** Extracted cubic sub volume from the labeled yellow dash line area in (**b–d**). **g** The right view of sub volume. **h** Representative ortho slices marked by black dash line in (**g**). **i** Volumes from segmentation by contrast corresponding to RuO₂ (blue) and interspace (black), respectively.

confirmed by the atomic line profiles corresponding line-scanning intensity (Fig. 3b) of the yellow lines 1–3 in Fig. 3a. In addition, the orientation of nanoparticles inside the crystal is randomly distributed along the [−111] zone axis (Ru and O atoms are arranged with a tetragonal structure), resulting in rich intra granular defects (Fig. S22) including the twinned structures shown in Fig. 3c (yellow dotted line box indicating the boundary). The fast Fourier transform (FFT) pattern of the corresponding region (1–5) in Fig. 3c further identifies that (110) and (101) crystal planes are arranged in different orientations (Fig. 3d).

The X-ray photoelectron spectroscopy (XPS) shows almost identical Ru 3*p* spectra to C-RuO₂ and K-RuO₂ (Fig. 3e), but two peaks of Ru 3$p_{3/2}$ and Ru 3$p_{1/2}$ for MD-RuO₂-BN shift about 0.3 eV to lower binding energy relative to C-RuO₂ and K-RuO₂, implying MD-RuO2-BN possesses more low charge Ru ions and oxygen vacancy defects (V_O). In O 1*s* spectra (Fig. 3f), the peak proportion attributable to V_O of MD-RuO₂-BN, C-RuO₂ and K-RuO₂ is 46.2%, 29.7%, and 31.8%, respectively, indicating an obvious increase of V_O concentration in MD-RuO₂-BN[15,18]. Besides, the binding energy position of the Ru-O characteristic peak for MD-RuO₂-BN also shifts by about 0.1 eV relative to C-RuO₂ and K-RuO₂, further suggesting a redistribution of charges. Moreover, MD-RuO₂-BN shows a stronger electron paramagnetic resonance (EPR) signal at *g* = 2.003, also proving that MD-RuO₂-BN contains more V_O than C-RuO₂ and K-RuO₂ (Fig. 3g).

The X-ray absorption near edge structure (XANES) of Ru *K*-edge for Ru powder, C-RuO₂ and MD-RuO₂-BN exhibits that the formation of Ru-O bond greatly pushes up the transition energy of XANES, and the absorption edge position for MD-RuO₂-BN is at lower energies compared with that of C-RuO₂ (Fig. 3h). These indicate that the average valence state of Ru in MD-RuO₂-BN is less than +4, also consistent with XPS analysis results. Corresponding extended X-ray absorption fine structure (EXAFS) spectra show that the average distance between Ru and O is lengthened and the strength of Ru-O bond is obviously weakened during the modification of pristine RuO₂ (Fig. 3i). Thus, the coordination environment of Ru is unsaturated, and the spatial fitting results (Fig. S23) further prove that the quantitative coordination number of Ru-O is reduced from 6.0 ± 0.4 to 4.4 ± 0.2. Finally, the

wavelet transform (WT) validates the analysis result from EXAFS spectra and visualizes the co-existence of Ru-O and Ru-Ru paths in MD-RuO₂-BN (Fig. 3j).

From the above investigations, we can find that there are multiscale defects including V_Ru, V_O, and intra/inter granular boundaries from point to surface in the ultrafine RuO₂ nanomonomers. These defects alter the local electronic structure of the catalyst by reducing oxidation state of Ru and lengthening Ru-O bond, and the coordination environment by generating generous low coordination Ru atoms, forming lattices-mismatched RuO₂. Both the lowered Ru valence state and weakened Ru-O interaction would inhibit the oxidation of lattice oxygen and the dissolution of high-valence Ru, resulting in greatly enhanced durability[20,46]. Therefore, the bicontinuous nanoreactors assembled from multiscale defective RuO₂ nanomonomers are expected to play an important role in catalysis.

### Electrochemical performance evaluation

We first tested the OER activity of MD-RuO₂-BN in 0.5 M H₂SO₄ (pH = 0.47, Fig. S24) with a three-electrode system. As shown in Fig. 4a, from polarization curves, it demonstrates a low overpotential of 196 @10 mA cm⁻² for MD-RuO₂-BN compared with that of K-RuO₂ (245 @10 mA cm⁻²) and C-RuO₂ (305 @10 mA cm⁻²), indicating the higher activity can be achieved by the structural design and defect engineering of RuO₂ nanoreactors. Moreover, we can find that MD-RuO₂-BN possesses the lower Tafel slope (Fig. 4b), smaller charge-transfer resistance (Fig. 4c) and higher electrochemically active surface area (Fig. S25) during OER processes. Meanwhile, the refinement of grains, the formation of bicontinuous structures, and the generation of various defects increase the surface energy of MD-RuO₂-BN (Fig. S26) and determine the strong hydrophilicity[47,48], resulting in the obviously decreased bubble contact angle of MD-RuO₂-BN (45°) compared with C-RuO₂ (60°), as shown in Fig. S27. In addition, the mass transfer is accelerated due to increased surface hydrophilicity and electronic structure optimization together[49]. Thus, the low OER overpotential of MD-RuO₂-BN can be attributed to the increase of the number of effective active sites, the intrinsic activity and mass transfer capability.

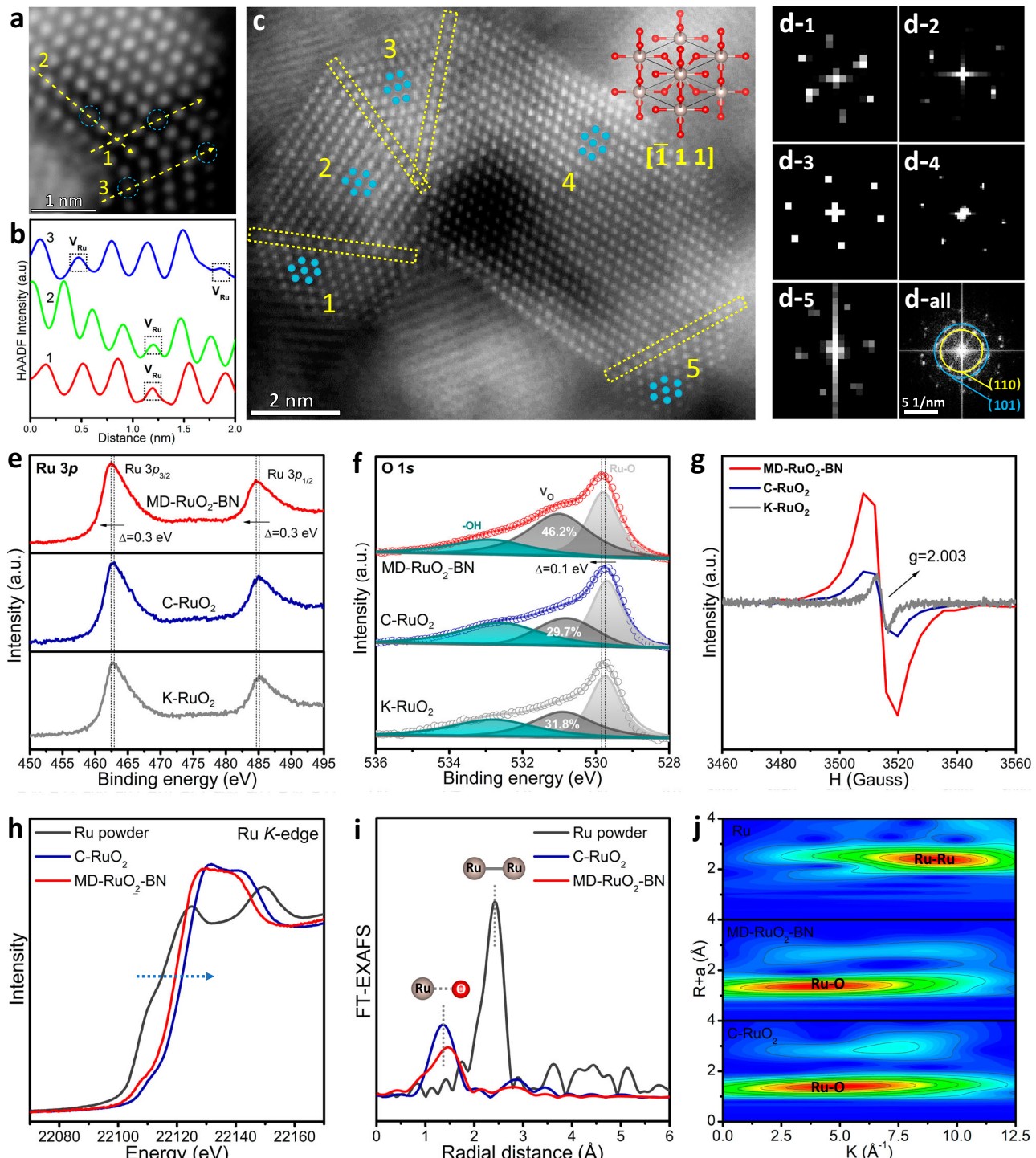

**Fig. 3 | Atomic and local electronic structures of catalysts. a** Atomic STEM image of MD-RuO₂-BN. **b** Corresponding line-scanning intensity profile of the yellow lines in (**a**). **c** Representative HAADT-STEM image of MD-RuO₂-BN. **d** Corresponding FFT images are obtained from the grains in (**c**). XPS spectra of Ru 3*p* (**e**) and O 1*s* (**f**) for MD-RuO₂-BN, K-RuO₂ and C-RuO₂. **g** EPR spectra of MD-RuO₂-BN, K-RuO₂ and C-RuO₂. **h** Ru *K*-edge XANES and (**i**) corresponding EXAFS spectra of Ru powder, MD-RuO₂-BN and C-RuO₂. **j** WT for the EXAFS signals.

Besides, the acid OER activity of MD-RuO₂-BN exceeds most other recently reported Ru-based catalysts (Table S1).

The accelerated degradation measurements show a slight shift of polarization curves for MD-RuO₂-BN before and after 1000 cycles in the OER process (Fig. S28). Furthermore, after 24 h continuous operation, the constant current chronopotentiometry (Fig. 4d) exhibits the increased OER potential by only 0.029 V (degradation rate of 1.2 mV h⁻¹), indicating the higher OER stability of MD-RuO₂-BN than

that of K-RuO₂ (5.3 mV h⁻¹) and C-RuO₂ (rapid degradation). A series of characterizations after durability test including XRD, XPS and STEM verify the phase and structural stability of MD-RuO₂-BN in acidic OER processes (Figs. S29 and 30). Moreover, the concentration of Ru ions in the electrolyte was measured using inductively coupled plasma-optical emission spectroscopy (ICP-OES). And the percentage of Ru dissolved from C-RuO₂ and K-RuO₂ during the OER is 5.6% and 4.5%, respectively, while it is only 2.9% for MD-RuO₂-BN. ICP-OES studies at regular

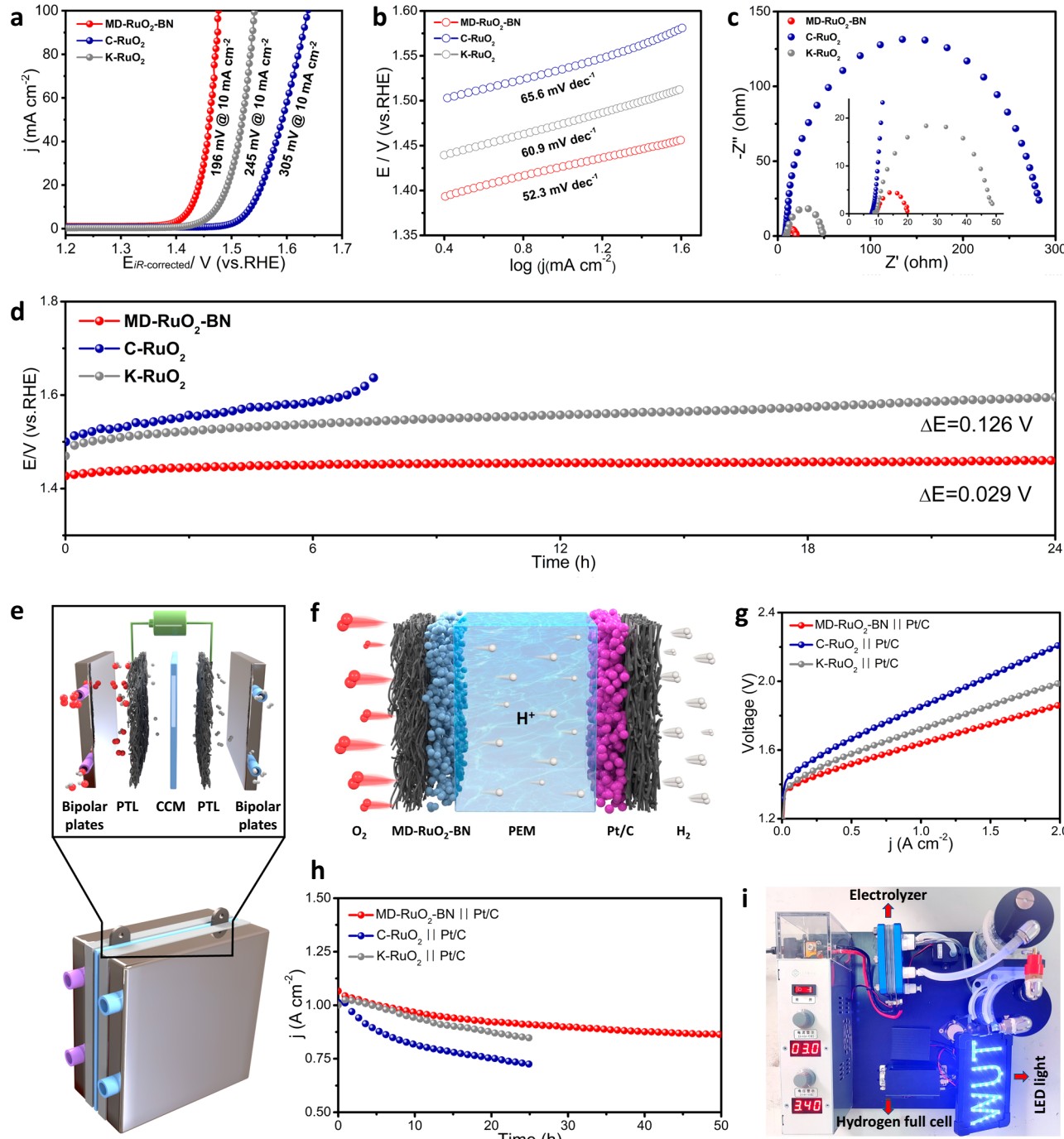

**Fig. 4 | Catalytic performance of MD-RuO$_2$-BN.** Polarization curves, Tafel curves and Nyquist plots of MD-RuO$_2$-BN, K-RuO$_2$ and C-RuO$_2$ (**a**–**c**). The high-frequency region of Nyquist plot is used to determine the solution resistance (about 9 Ω) for *iR*-correction. **d** Constant current stability testing for OER, no *iR*-correction. Schematic diagram of a typical PEMWE device (**e**) and the MEA components (**f**). Polarization curves (**g**) and time-dependent current density curves (**h**) of the PEM electrolyzer measured at 80 °C, no *iR*-correction. **i** Demonstration of hydrogen production from PEMWE to drive the hydrogen fuel cell.

intervals (24 h) while running extended chronopotentiometry experiments for 7 days (Figs. S31 and 32) further indicate effective inhibition of the dissolution of Ru in MD-RuO$_2$-BN compared with K-RuO$_2$ (Table S2). Thus, it is clear that MD-RuO$_2$-BN possesses an excellent balance between activity and stability for OER.

A H-type electrolytic cell (MD-RuO$_2$-BN ‖ Pt/C) was assembled for acidic overall water splitting, and only 1.465 V is needed to drive the current density of 10 mA cm$^{-2}$ (Fig. S33). Such performance is far better than commercial electrode pairs (C-RuO$_2$ ‖ Pt/C, 1.564 V at 10 mA cm$^{-2}$). The water drainage method demonstrates the almost 100% Faradaic yield of MD-RuO$_2$-BN ‖ Pt/C systems (Figs. S34 and 35)[50]. Therefore, a PEMWE single cell consisting of catalyst coated membrane (CCM), porous transport layers (PTL) and bipolar plates was finally installed (Fig. 4e, f) and tested (Fig. S36). The catalyst at anode and cathode is MD-RuO$_2$-BN and commercial 40% Pt/C, respectively, and the PEM is Nafion membrane (Nafion 115®). Surface stress distribution nephogram of PTL and pressure-sensitive paper suggests the good contact of the catalytic layers with PTL under an assembly pressure of 8 N m (Fig. S37). Besides, the cross-sectional and planar SEM images of the CCM with MD-RuO$_2$-BN (Fig. S38) show that the

catalytic layer is uniformly coated on the membrane, and the thickness of catalytic layer is ~20 μm.

The polarization curve (Fig. 4g) of PEMWEs shows that, to reach a current density of 1 A cm$^{-2}$ for water electrolysis, MD-RuO$_2$-BN-PEM-Pt/C with MD-RuO$_2$-BN as anode catalyst only requires a cell voltage of 1.64 V, superior to those of K-RuO$_2$-PEM-Pt/C (1.72 V@1 A cm$^{-2}$) and C-RuO$_2$-PEM-Pt/C (1.85 V@1 A cm$^{-2}$). Besides, we selected 1.65, 1.72, and 1.85 V as the constant voltage for stability tests, which should yield ~1 A/cm$^2$ based on the LSV. Time-dependent current density curves (Fig. 4h) reveal that our MD-RuO$_2$-BN-based electrolyzer well maintains water electrolysis activity for 50 h, while both K-RuO$_2$ and C-RuO$_2$ experiences clear decline within only 25 h of operation, especially C-RuO$_2$. This further proves the improved stability of our designed RuO$_2$ catalysts. Moreover, Fig. S39 shows that there is no detachment of the MD-RuO$_2$-BN catalytic layer on anode after operation at 1 A cm$^{-2}$ for 50 h. And the cross-sectional and planar morphology of the MEA further demonstrate that the catalytic layer and membrane structure are well preserved (Fig. S40). Finally, we designed a regenerative fuel cell system combining self-made electrolyzer and commercial hydrogen fuel cells (Figs. 4i and S41 and Supplementary Movie 5). The illumination of the light emitting diode marks the realization of green recycling for hydrogen coming from the water and then returning to water.

## Mechanism analysis of enhanced activity and stability

To explore the mechanism for enhanced OER performance of RuO$_2$ with the assistance of oxygen vacancies (V$_O$), Ru vacancies (V$_{Ru}$), and twin boundaries (T), the DFT calculations were conducted. We first modeled a series of RuO$_2$ slabs (Fig. 5a) to investigate the adsorption of critical intermediates on routine Ru sites (Fig. S42) for pure RuO$_2$, RuO$_2$-V$_O$, RuO$_2$-V$_{Ru}$, RuO$_2$-T, and RuO$_2$ with both V$_O$, V$_{Ru}$ and T (RuO$_2$-T-V$_{Ru,O}$). As displayed in Fig. S43, the rate-determining step (RDS) of OER on this Ru site for RuO$_2$-T-V$_{Ru,O}$ is not optimized, inconsistent with the experimental results. Besides routine Ru sites, further investigation (Fig. S44) unveils that the OER activity is obviously increased on twin boundary Ru sites (TB-Ru) of RuO$_2$-T-V$_{Ru,O}$. Specifically, pure RuO$_2$, RuO$_2$-T-V$_O$, and RuO$_2$-T-V$_{Ru}$ show overpotentials of 0.70, 0.79 and 0.69 V, respectively, in the O$_2$ desorption or the OOH* formation step, while RuO$_2$-T-V$_{Ru,O}$ exhibits a low overpotential of 0.22 V in the OOH*-to-O$_2$ converting step (Fig. 5b). Therefore, the high OER activity is attributed to the significant decrease in the kinetic energy barrier caused by the TB-Ru site. In addition, given that the strong adsorption of oxygenated intermediates usually induces the instability of metal sites[51], RuO$_2$-T-V$_{Ru,O}$ shows the increased $\Delta G_{O^*}$ and $\Delta G_{OH^*}$ relative to pure RuO$_2$, leading to a weaker adsorption and thus improvement of stability (Fig. 5c). In sum, a water oxidation mechanism for MD-RuO$_2$-BN is proposed based on experimental and computational analysis (Fig. 5d). The synergistic effect of multi-scale defects including V$_O$, V$_{Ru}$ and T highly facilitates OER on T-Ru sites.

The in situ Raman spectra were further performed to probe the surface state of catalysts under constant potentials progressively stepping to positive limits from the open-circuit potential (Fig. S45). As shown in Fig. 5e, f, the three major Raman features of rutile RuO$_2$, namely the $E_g$ (518 cm$^{-1}$), $A_{1g}$ (642 cm$^{-1}$) and $B_{2g}$ (702 cm$^{-1}$) vibration modes can be observed both on MD-RuO$_2$-BN and C-RuO$_2$ in ordinary 0.5 M H$_2$SO$_4$. Besides, two Raman bands at about 430 and 588 cm$^{-1}$ assign to Ru$^{4+}$-O bonds and Ru$^{3+}$-O bonds, respectively[52]. Significantly, when further normalizing the intensity of the band at 588 and 430 cm$^{-1}$ (Fig. 5g), we can find that MD-RuO$_2$-BN presents a higher intensity ratio (about 1.0) than C-RuO$_2$ (about 0.4), thereby possessing more Ru$^{3+}$ species on the surface, further supporting the higher content of low-valent Ru species on the MD-RuO$_2$-BN surface caused by multiscale defects. The lowered Ru valence state would inhibit the dissolution of high-valence Ru, resulting in enhanced OER durability.

We also carried out in situ Raman in H$_2^{18}$O labeled 0.5 M H$_2$SO$_4$ (Fig. 5h, i). By comparing the shift of the characteristic vibration modes

attributed to RuO$_2$ in ordinary and H$_2^{18}$O labeled electrolytes, it can be found that C-RuO$_2$ exhibits a certain degree of negative displacement at $E_g$, $A_{1g}$ and $B_{2g}$ (Fig. 5j). This is mainly owing to the exchange of lattice oxygen in RuO$_2$ with $^{18}$O in the electrolyte at an applied potential, resulting in a change in vibration frequency caused by partial replacement of lattice Ru-$^{16}$O with Ru-$^{18}$O[30]. While no isotope effect is observed in MD-RuO$_2$-BN due to reduced oxidation state of Ru, and the weakened Ru-O interaction suppresses the participation of lattice oxygen. The above results provide insights into the stability improvement of MD-RuO$_2$-BN from the changes of the in situ surface structure.

## Discussion

In summary, we assemble RuO$_2$ nanomonomers with abundant Ru/O vacancies and intra/inter granular boundaries into a bicontinuous nanoreactor, achieving the simultaneous optimization of electron- and micro-structures for RuO$_2$. The reduced oxidation state of Ru and the extended Ru-O bond weaken the Ru-O interaction, and inhibit the oxidation of lattice oxygen and the dissolution of high-valence Ru, thereby enhancing durability of RuO$_2$. Meanwhile, abundant reaction regions and efficient low coordination active atoms jointly enhance catalytic activity. As a result, the customized MD-RuO$_2$-BN exhibits unparallel oxygen evolution reaction (OER) activity and stability in three electrode cell setup and proton-exchange membrane water electrolyzers (PEMWEs). Furthermore, the demonstration of the integrated hydrogen-water circulating power supply system provides more opportunities for application of Ru-based materials. This work shews an insight into improving catalytic performance of the Ir-free-based OER catalysts, and will stimulate the development of PEMWEs for large-scale green H$_2$ generation.

## Methods

### Chemicals

Ruthenium (IV) oxide monohydrate (RuO$_2$·H$_2$O) was purchased from Wokai Reagents Ltd. Anhydrous lithium chloride (LiCl) and potassium chloride (KCl), sulfuric acid (H$_2$SO$_4$) and isopropyl alcohol were purchased from Sinopharm Chemical Reagent Co., Ltd. Commercial RuO$_2$, Pt/C (20 wt%) and Nafion (5 wt%) were obtained from Sigma-Aldrich. H$_2^{18}$O was purchased from Rhawn Reagents Ltd. All the reagents are analytical grade and used without further treatment. Deionized (DI) water was employed as solvent.

### Material syntheses

The KCl-LiCl molten salt assisted system was used for the facile synthesis of the target catalyst MD-RuO$_2$-BN. Specifically, 1.35 g KCl, 1.15 g LiCl and 0.15 g RuO$_2$·H$_2$O powder were uniformly ground into a mixture under conditions of isolating oxygen and water, waiting for the subsequent high-temperature reaction. The reaction condition was to maintain 500 °C for 4 h under inert gas to ensure sufficient reaction. After the reaction was completed, it was necessary to thoroughly wash the solid product with deionized water to remove residual KCl-LiCl. Finally, after vacuum dried at 60 °C overnight, the target catalyst MD-RuO$_2$-BN was obtained. We also carried out a blank experiment using KCl instead of KCl-LiCl eutectic system as control under the same conditions (K-RuO$_2$).

### Characterization

X-ray diffraction (XRD) patterns were collected on a Rigaku X-ray diffractometer equipped with a Cu $K\alpha$ radiation source to obtain the crystalline structure of all samples. Nitrogen adsorption/desorption isotherms were measured on ASAP2020M apparatus to obtain the Brunauer-Emmett-Teller (BET) surface area and Barrett-Joyner-Halenda pore diameter. X-ray photoelectron spectroscopy (XPS), electron paramagnetic resonance (EPR) and synchrotron radiation X-ray absorption spectroscopy (XAS) were carried out to reveal the electronic structure and valence bond structure. The morphology and

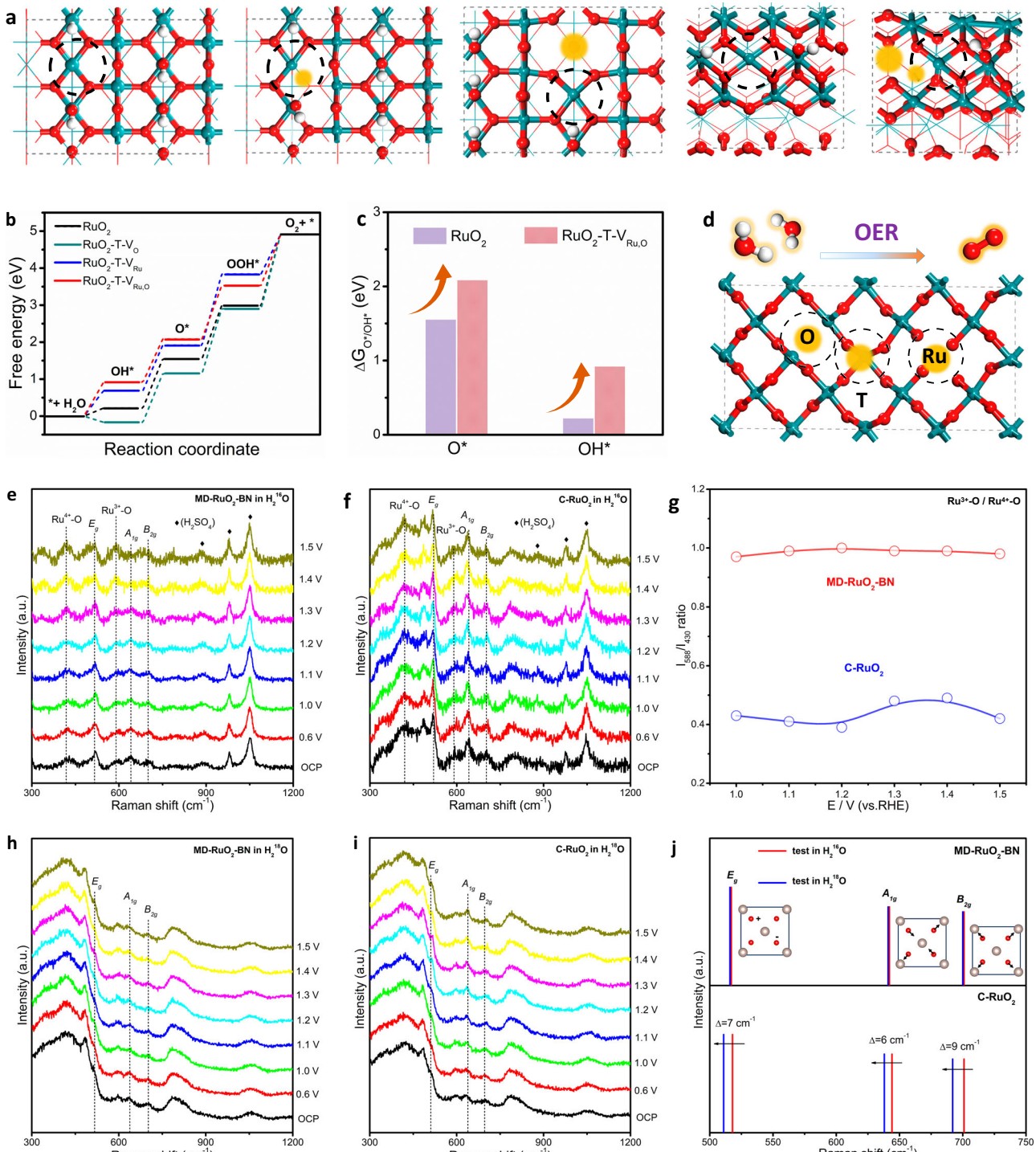

**Fig. 5 | DFT calculations and in situ Raman spectra. a** Top views of pure RuO₂, RuO₂-V_O, RuO₂-V_Ru, RuO₂-T, and RuO₂-T-V_Ru, O. The red and indigo balls represent the O and Ru atoms, respectively. The black circles and yellow highlights denote the active sites and vacancy, respectively. **b** The free energy profile of OER on TB-Ru sites. **c** Calculated O* and OH* adsorption energy of RuO₂ and RuO₂-T-V_Ru, O. **d** The active sites (yellow highlights) on RuO₂-T-V_Ru, O for catalyzing OER. **e, f** Raman spectra for MD-RuO₂-BN and C-RuO₂ in ordinary 0.5 M H₂SO₄. **g** Normalized intensity of Raman band at 588 cm⁻¹ to that at 430 cm⁻¹ on the catalysts as a function of applied potential. **h, i** Raman spectra for MD-RuO₂-BN and C-RuO₂ in H₂¹⁸O labeled 0.5 M H₂SO₄. **j** The shifting of Raman peaks corresponding to the E_g, A₁_g and B₂_g vibration modes for MD-RuO₂-BN and C-RuO₂.

structure were characterized by double spherical aberration-corrected scanning transmission electron microscope (AC-STEM, Titan Cubed Themis G2 300). Inductively coupled plasma-optical emission spectroscopy (ICP-OES) was carried (700 Series, Agilent Technologies) for the leaching measurements. For in situ Raman measurements, Raman spectra were obtained by Horiba LabRAM HR Evolution with a He/Ne laser of $\lambda = 532$ nm. The multi-potential chronoamperometry test was performed by an electrochemical workstation (Autolab PGSTAT 204) in a customized Teflon cell with a 0.5 M H₂SO₄ electrolyte. A catalyst-supported gold electrode (diameter = 0.3 cm) worked as the working electrode, a saturated calomel electrode as the reference electrode, and a polished platinum wire as the counter electrode.

## Electrochemical measurements

All electrochemical measurements were performed in a conventional three-electrode system at room temperature using a CHI 660E electrochemical analyzer (CHI Instruments, Shanghai, China). The acidic ($0.5 M H_2SO_4$) electrochemical measurements were performed using a saturated calomel electrode (SCE) as the reference electrode, a graphite plate as the counter electrode, and a glassy carbon electrode with a diameter of 3 mm as the working electrode. The catalyst ink was prepared by dispersing 5 mg as-prepared sample into a mixture (900 μl isopropyl alcohol, 80 μl water and 20 μl 5% Nafion solution) and ultrasonic dispersion for 30 min. For comparison, 5 mg commercial catalyst powder ($RuO_2$) was evenly dispersed into the same mixture. The final loading for all catalysts on the glassy carbon was about $0.7 mg cm^{-2}$. Polarization data were obtained at a scan rate of $5 mV s^{-1}$. In this work, all potentials measured against SCE were converted to the reversible hydrogen electrode (RHE) scale using:

$$E(\text{vs RHE}) = E(\text{vs SCE}) + 0.241 V + 0.0591 \times pH \quad (1)$$

In the given equation, 0.241 V was obtained by calibration with respect to the RHE and the pH value of the electrolyte was determined to be 0.47 by several measurements.

All polarization curves were *iR*-corrected:

$$E_{iR-\text{corrected}} = E(\text{vs RHE}) - iR \quad (2)$$

where the *R* is the solution resistance, and the high-frequency region of Nyquist plot is used to determine the *R* (about 9 Ω) for *iR*-correction. The electrochemical impedance spectroscopy (EIS) was conducted at the corresponding potentials of $10 mA cm^{-2}$ from LSV curves, with the frequency range of 0.01 Hz to 100 kHz with AC amplitude of 10 mV. The electrochemical double layer capacitance ($C_{dl}$) was determined with typical cyclic voltammetry (CV) measurements at various scan rates (20, 40, 60, 80 and $100 mV s^{-1}$) in nonreactive region. The durability was evaluated by accelerated degradation measurements and constant current chronopotentiometry. The obtained electrocatalyst and Pt/C were used as anode and cathode in a two-electrode configuration for overall water splitting. And the generated $H_2$ and $O_2$ gases during overall water splitting were quantitatively collected by the water drainage method for evaluating Faraday efficiency.

## PEMWE tests

OER activities of MD-RuO₂-BN, K-RuO₂ and C-RuO₂ in practical applications were evaluated in home-made PEMWE single cell which consisted of CCM, PTL and bipolar plates. First, the catalyst layers were prepared. The cathode catalyst ink was prepared by mixing 35 mg Pt/C (40 wt%) powder, 300 mg of Nafion solution (5 wt%), 2 ml of DI water and 8 ml of isopropanol and then sonicated for 60 min in an ice bath. The anode catalyst ink was prepared by mixing 40 mg of RuO₂ powder, 200 mg of Nafion solution (5 wt%), 1 ml of DI water and 4 ml of isopropanol and then sonicated for 60 min in an ice bath. The catalyst inks were transferred to sheets of polytetrafluoroethylene (PTFE) by spraying, forming catalyst layers. Next, we performed the assembly of CCM, where the electrolyte selected is Nafion 115® membrane (127 μm). The Nafion 115® membrane requires the following pretreatment: 5% hydrogen peroxide was treated at 80 degrees for 1 h, and then soaked in deionized water for 0.5 h; 5% dilute sulfuric acid was boiled at 80°C for 1 h, and then soaked in deionized water for 0.5 h. Then, the catalytic layers loaded on PTFE sheets were transferred to both sides of the Nafion 115® membrane by the traditional decal method or the thermal transfer printing[53]. The specific operating parameters were: hot-pressing temperature of 130 °C, pressure of 20 MPa, and hot-pressing time of 10 min. To prevent deformation and collapse of CCM caused by shrinkage stress during the cooling process, after hot-pressing, it was necessary to use a flat heavy object to press for 1 min and then carefully peel off the surface PTFE. And then the CCM with the active area of 4 cm² (the loading of the cathode and anode catalyst layer was $2 mg cm^{-2}$ and $1 mg cm^{-2}$, respectively) was obtained. Finally, a torque wrench (8 N m) was used to assemble the bipolar plates, PTL (the titanium felt with a thickness of 0.6 mm), and prepared CCM into a PEMWE single cell, and relevant performance tests (polarization curves, time-dependent current density curves) were conducted in the testing system (operating temperature of 80 °C, flow rate of $30 ml min^{-1}$).

## DFT calculations

The DFT calculations method performed in this work using Vienna ab initio simulation package (VASP)[54,55]. Specifically, the kinetic energy with a cut-off energy of 450 eV was set for each atom within RuO₂ slabs. For electronic structure, *k*-point sampling with a ($1 \times 1 \times 1$) mesh within the Monkhorst-Pack scheme was utilized. And during the optimization, we employed Gaussian smearing (ISMEAR = 0) with a smearing width of 0.02 eV (SIGMA = 0.02). We modeled the valence electron in the form of -H (ultrasoft test, $1s^1$), O ($2s^2$, $2p^4$), Ru ($4p^6$, $4d^7$, $5s^1$). The vacuum layer of 15 Å was applied to avoid lateral interactions. For the calculation of the change in the Gibbs free energy ($\Delta G$), the classic four-electron OER steps were adopted:

$$H_2O + * = OH^* + H^+ + e^- \quad (3)$$

$$OH^* = O^* + H^+ + e^- \quad (4)$$

$$O^* + H_2O = OOH^* + H^+ + e^- \quad (5)$$

$$OOH^* = O_2 + H^+ + e^- \quad (6)$$

The $\Delta G$ value of each elemental OER steps was calculated based on the following equation:

$$\Delta G = \Delta E + \Delta E_{ZPE} - T\Delta S \quad (7)$$

where $\Delta E$, $\Delta E_{ZPE}$ and $\Delta S$ values represent the energy change, the difference on energy, and the entropy between the adsorbed state and gas, respectively. *T* is generally 298.15 K.

# Data availability

The data that support the plots are available within this paper and its Supplementary Information. All other relevant data that support the findings of this study are available from the corresponding authors on reasonable request. Figures 1, 3–5 and S4, S14, S23, S25, S26, S28, S29, S32, S33, S35, S43 data generated in this study are provided in the Source data files. Source data are provided with this paper.

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

## Acknowledgements

This work was supported by the National Natural Science Foundation of China (Grant Nos. 22075223, 22379117, S.M.), the State Key Laboratory of Advanced Technology for Materials Synthesis and Processing (Wuhan University of Technology) (2023-ZT-1, S.M.).

## Author contributions

D.C. and S.M. conceived and designed the studies. D.C. synthesized the materials, performed their electrochemical properties, analyzed the data and wrote the paper. R.Y. and K.Y. supported the characterizations and analysis. R.L. performed the DFT calculations and result analyses. H.Z., J.J. and Y.Y. contributed to the materials synthesis and electrochemical measurements. J.Z., J.W. and S.M. provided helpful suggestions and revised the manuscript. All authors discussed the results and critically reviewed the manuscript.

## Competing interests

The authors declare no competing interests.
