## [Peer Review File · Nature Communications]

REVIEWER COMMENTS

Reviewer #1 (Remarks to the Author):

The manuscript is well presented and addresses the important problem in PEM electrolysis of replacing IrO₂ OER catalysts. The authors propose an interesting approach using eutectic KCl-LiCl to create a crystalline RuO₂ product. This product demonstrated good stability when tested in a small-scale flow electrolytic cell (PEMWE), when benchmarked against commercial RuO₂. As expected, the MD-RuO₂-BN outperformed commercial RuO₂, likely due to its higher surface area and mesoporosity, consistent with N₂ adsorption measurements. Intriguingly, despite the significantly higher surface area, MD-RuO₂-BN maintained a current of 0.2 A/cm², while the C-RuO₂ faded quickly. In principle, this work shows promise for publication. However, some significant issues remain to be addressed before the work would be accepted, as listed below:

1. The authors elaborate a theory that they have created some kind of nanoreactors. This may be the case, but in reality, the TEM results in Figure 1c suggest these structures are just agglomerations of nanoparticles. The overlap between the nanoparticles creates mesopores, consistent with the N₂ adsorption isotherm results in Fig. 1f. There is nothing particularly special about these agglomerate structures, as they are quite common. I am not convinced the tomography results validate calling these agglomerates "nanoreactors."
2. The synthetic route is very interesting, and the outcome looks promising. However, with only a single sample prepared, the results are not yet convincing. The benchmark used is C-RuO₂. If the authors are trying to prove that using a eutectic solution is a superior method, they must demonstrate that a RuO₂*H₂O product made in KCl alone is of inferior quality to MD-RuO₂-BN. I am concerned that the authors have built a massive theory and explanation based on just a SINGLE sample. Please run a blank experiment using KCl instead of KCl-LiCl as a control. Please critically evaluate your data and be honest about the limitations before overselling your paper based on a single sample. More replicates (which are prepared under similar conditions) are needed to substantiate the claims.
3. Looking at the results in Figure 3, the difference between MD-RuO₂-BN and C-RuO₂ seems negligible. However, as mentioned in point 2, these two products are very different in preparation. Since the authors claim a unique ability of the eutectic solution to create a superior catalyst, it is important to compare to a blank sample prepared using just KCl. Despite the samples C-RuO₂ and MD-RuO₂-BN are different, unlike the authors, I have doubts there is an appreciable change in Ru oxidation state, and if there was, the highly oxidative conditions at the anode would likely oxidize Ru to a high state anyway. Can the authors explain their conclusions in the context of actual processes likely to occur at the anode? If not, it is perfectly acceptable to avoid confusing the readers (given the general readership of Nature Communications), with terminology like "unique bi-continuous nanoreactors" when the evidence is lacking.
4. What was the dissolution rate of the control C-RuO₂ in H₂SO₄ according to ICP? What would be the dissolution rate of the control RuO₂ prepared in KCl?

5. It is fine to carry out tests at negligibly low current densities of 10 mA / cm² in sulfuric acid as it allows for the comparison with literature. However, authors provided no details (also not in the SI) about the testing conditions. What was the size of glassy carbon electrode used, for example? Please provide comprehensive information and also run the experiment on a blank prepared in KCl.

6. My primary concern is the performance in the flow-cell PEMWE electrolyser (Fig. 4g). The MD-RuO₂-BN reaches 1A/cm² at a relatively low potential (below 2V), as expected given its high surface area and small RuO₂ nanoparticles. However, the high potential seemingly required for C-RuO₂ is unexpected. We routinely reach 1A/cm² below 2V by running a LSV with commercial RuO₂ in my lab (including those immobilised on Ti GDL by spraying). The LSV curve for RuO₂ does not seem correct - there were likely serious issues in preparation.

There could be issues with the electrode preparation and spraying process, which the authors describe insufficiently. Please provide comprehensive details on the preparation methods. The authors should re-run tests with C-RuO₂ and RuO₂ in KCl for comparison. Please update both methods and preparation sections both in the main text and SI. SEM imaging of the RuO₂ immobilized on Ti GDL would also be more useful than the 3-electrode system work. Overall, more information is needed on the electrode preparation and benchmarking experiments. As mentioned below the stability of MD-RuO₂-BN is important achievement but given the issues with benchmark and unusual LSV for C-RuO₂ more experiments (and description of the experiments) are required.

7. Figure 4h shows MD-RuO₂-BN has good stability at 0.2 A/cm², but the applied potential is not reported. Please rerun the test at 2V (which should yield ~1 A/cm² based on the LSV), as this is a realistic potential used in good performing electrolyzers.

Again, it is important to run a similar stability test on RuO₂ prepared in KCl alone for comparison. Again, the fading of C-RuO₂ at such a quick rate in just 0.2 A/cm² makes me concerned about preparation of the electrodes. More work is needed here as this seems to be the key to application of MD-RuO₂-BN.

8. The isotope exchange results are quite interesting, but as mentioned above, have limited relevance from an applications perspective since the 3-electrode testing was done at incredibly low current densities of 10 mA/cm². This testing provides little insight into how the material would perform under actual PEM electrolysis conditions. It would be more useful to provide photos and SEM images of the electrodes after PEMWE testing, rather than spending so much effort on highly complicated 3-electrode experiments at non-representative current densities. While academically interesting, the priority should be characterizing performance at industrially-relevant currents in a PEMWE cell, rather than low current density 3-electrode testing.

There are some minor issues that can be covered after the authors respond to key issues above and in particular run a control experiment on RuO₂ prepared in KCl.

Overall, this is a promising paper, but requires some additional work before it would be acceptable for publication. The authors have an interesting approach using the eutectic KCl-LiCl solution to synthesize the MD-RuO₂-BN material. However, more evidence is needed to demonstrate superiority over benchmark RuO₂, particularly testing under industrially-relevant conditions in a PEMWE cell. The suggestions provided, such as comprehensively detailing the preparation methods, performing control

syntheses in KCl, testing at higher currents and potentials, and providing post-PEMWE electrode imaging, would strengthen the claims and conclusions. Addressing these points should improve the manuscript and likelihood of publication.

Reviewer #2 (Remarks to the Author):

The author used liquid molten salt to trigger the dual modulation of Ru electronic characteristics and local microenvironments, and successfully synthesized a heterogeneous multiscale defective RuO₂ “bicontinuous nanoreactors” catalyst. The synthesized MD-RuO₂-BN exhibited a better alkaline oxygen evolution performance than the commercial RuO₂ (C-RuO₂). Specifically, RuO₂-BN delivered an overpotential of 196 mV (vs RHE) at a current density of 10 mA cm⁻² in electrochemical tests and 2.02 V at a current density of 1 A cm⁻² in PEMWE devices. We appreciate the authors’ efforts in this research. However, there are many contradictions in the manuscript and the lack of innovations. Some similar works have been published previously (Nat. Commun. 13, 5716 (2022); Nat. Commun. 14, 1412 (2023)). The above issues prevent us from recommending the manuscript for publication in Nature Communications.

1. The synthesis mechanism of the “bicontinuous nanoreactors” needed to be elaborated in detail
2. Why named the MD-RuO₂-BN “bicontinuous nanoreactors”? In fact, many kinds of RuO₂ have a pore structure, could they all be categorized as nanoreactors? (Chinese Journal of Chemical Engineering 55 (2023) 93-10)
3. In the XRD patterns of MD-RuO₂-BN and C-RuO₂ (Figure 1e), the diffraction peak of MD-RuO₂ was weaker than that of C-RuO₂ obviously.
4. In the Ru 3p XPS spectra of MD-RuO₂-BN and C-RuO₂ (Figure 3e), two peaks assigned to Ru 3p of C-RuO₂ were located at higher energy than that of MD-RuO₂-BN, indicating a higher Ru valence state in C-RuO₂ than MD-RuO₂-BN. However, the author's description here was “C-RuO₂ shift about 0.3 eV to lower binding energy relative to the obtained MD-RuO₂-BN”. In addition, previous studies revealed that Ru species with a higher oxidation state could enhance their OER activity. Why do the lower valence states of Ru species have better OER performance in this manuscript? (Nat. Mater. 22, 100-108 (2023)).
5. Due to the different half-peak widths of the characteristic peaks attributed to oxygen vacancies in MD-RuO₂-BN and C-RuO₂, it is not possible to determine the relative concentration of oxygen vacancies in the two catalysts from the height of the peaks. The authors should give the ratio of the different characteristic peaks to the whole peak to determine the concentration of oxygen vacancies (Figure 3f). In addition, since the valence state of Ru species in MD-RuO₂-BN has changed, the binding energy position of the Ru-O characteristic peak here should also change.
6. Both MD-RuO₂-BN and C-RuO₂ have signals attributed to oxygen vacancies in their EPR spectra, and the stronger signal of MD-RuO₂-BN indicated that it contains more oxygen vacancies. The author's statement "offers direct evidence for the existence of VO" should be replaced with the statement that “MD-RuO₂-BN contained more oxygen vacancies than C-RuO₂”.
7. Alterations in the Ru valence state of MD-RuO₂-BN and C-RuO₂ needed to be differentiated on the shift of the absorption edge rather than the white line peak intensity (Figure 3h) (Nat. Commun. 14, 2517 (2023); Nat. Catal. 2, 304-313 (2019)).

8. What is the reason for the increased hydrophilicity of the MD-RuO₂-BN surface?
9. In the theoretical calculations, what exactly are the sites that enhance the performance of OER, is it the grain boundaries or the reduced particle size, or the presence of vacancies that play a dominant role?
10. There are a large number of grammatical errors and data results analysis errors in the manuscript. Authors need to check the manuscript carefully.

Response to Reviewer #1

The manuscript is well presented and addresses the important problem in PEM electrolysis of replacing IrO₂ OER catalysts. The authors propose an interesting approach using eutectic KCl-LiCl to create a crystalline RuO₂ product. This product demonstrated good stability when tested in a small-scale flow electrolytic cell (PEMWE), when benchmarked against commercial RuO₂. As expected, the MD-RuO₂-BN outperformed commercial RuO₂, likely due to its higher surface area and mesoporosity, consistent with N₂ adsorption measurements. Intriguingly, despite the significantly higher surface area, MD-RuO₂-BN maintained a current of 0.2 A/cm², while the C-RuO₂ faded quickly. In principle, this work shows promise for publication. However, some significant issues remain to be addressed before the work would be accepted, as listed below:

Reply: We really appreciate your recognition and advices on our manuscript. Overall, based on your constructive comments, in revised manuscript, we have further clarified comprehensive and detailed preparation methods, supplemented the performing control syntheses and characterization in KCl, improved the testing under industrially-relevant conditions in a PEMWE cell, and provided post-PEMWE electrode imaging, to strengthen the claims and conclusions of this article. Through resolving these issues, we hope that the revised version can receive your continued favorable consideration. And the specific response for each comment is attached below.

Q1. The authors elaborate a theory that they have created some kind of nanoreactors. This may be the case, but in reality, the TEM results in Figure 1c suggest these structures are just agglomerations of nanoparticles. The overlap between the nanoparticles creates mesopores, consistent with the N₂ adsorption isotherm results in Fig. 1f. There is nothing particularly special about these agglomerate structures, as they are quite common. I am not convinced the tomography results validate calling these agglomerates "nanoreactors."

Reply: Thanks for your valuable comments. Herein, the prepared bicontinuous structure that we refer to consists of numerous ultrafine RuO₂ monomers with a particle

size of approximately 3nm and internal pores as the nanoreactor (**Figure 1c, Figure 2**). Of course, not all materials with pore structures can be called nanoreactors. Next, we will discuss in detail for the basic requirements that nanoreactors should meet.

First of all, the nanoreactor refers to a mesoscopic environment where chemical reactions are limited by the nanoscale space (*Angew. Chem. Int. Ed.* **2022**, *61*, e202204371; *Adv. Funct. Mater.* **2022**, *32*, 2205569). Therefore, ultrafine (close to the sub nanometer scale), uniform, and continuous monomers are crucial conditions for forming a nanoreactor. Obviously, many agglomerate structures do not meet this condition. Moreover, the nanoreactor should possess a periodically regular porous structure with three-dimensional interconnections, which is crucial for increasing the area of charge transfer interfaces, enriching active sites, and improving mass transfer efficiency (*Adv. Energy Mater.* **2020**, *10*, 2000651; *Angew. Chem. Int. Ed.* **2023**, *62*, e202300478). If the particles are too large and uneven or the pore structure is dispersed and not connected with each other, it is difficult to form a cavity with an enrichment effect, thus slowing down mass transfer and reaction kinetics.

For example, in this article, the microstructure of MD-RuO₂-BN (**Figure R1a, b**) and C-RuO₂ as commercial RuO₂ (**Figure R1c, d**) observed by TEM shows that the RuO₂ unit in MD-RuO₂-BN is smaller and uniform, while the C-RuO₂ possesses larger and uneven grain size, with a disordered spatial structure and obvious agglomeration. Therefore, by further combining advanced three-dimensional tomograph reconstruction technology (**Figure 2**), we obtain the unique RuO₂ catalyst (MD-RuO₂-BN) with a special structure as a nanoreactor.

It is worth noting that the liquid molten salt formed by KCl-LiCl eutectic at high temperatures is an important medium for the formation of the RuO₂ nanoreactor. According to your important suggestion, we have added the nanostructure characterization (**Figure R2**) of RuO₂ synthesized under the same conditions in the presence of only KCl (K-RuO₂). Here, due to the much higher melting point of KCl

(770 °C) than the synthesis temperature (500 °C), the reaction system will not be able to produce a liquid reaction medium. By comparing **Figure R1** and **Figure R2**, it can be found that the average grain size of K-RuO₂ (~10 nm) is relatively reduced to C-RuO₂ (~20 nm), but it is still significantly larger than MD-RuO₂-BN (~3 nm). Importantly, due to the lack of a liquid medium to provide a uniform growth environment, the grain size of K-RuO₂ is uneven and disorderly arranged, making it difficult to meet the basic requirements for forming nanoreactors mentioned above.

Moreover, 70 STEM-HAADF images for tomography reconstruction on the obtained K-RuO₂ were collected by a 1-2° interval (**Figure R3**). The resulting reconstructed structural unit is displayed in **Figure R4**. The sub volume extraction and segmentation (**Figure R5**) indicate that K-RuO₂ does not possess a nanoreactor structure like in MD-RuO₂-BN, further confirming our idea about nanoreactors that we describe.

In addition, we have made a corresponding revision in the manuscript, emphasizing the basic requirements that nanoreactors should meet. And the relevant figures have been added in the supporting information. The revision is shown as below:

(Page 9-10) “In addition, to demonstrate the role of eutectic solutions on creating the unique bicontinuous structured nanoreactor and further analyze the formation mechanism, we conducted a control experiment using KCl instead of the KCl-LiCl eutectic system under the same condition (K-RuO₂, Figure S14). Here, due to the higher melting point of KCl (770 °C) than the synthesis temperature (500 °C), the reaction system is unable to produce a liquid reaction medium. Corresponding nanostructure characterizations (Figure S15-16) show that the average grain size of K-RuO₂ (~10 nm) is reduced relative to C-RuO₂ (~20 nm), but still significantly larger than that of MD-RuO₂-BN (~3 nm). Importantly, due to the lack of a liquid medium to provide a uniform growth environment, the grain size of K-RuO₂ is uneven and disorderly arranged. Moreover, the 3D tomography reconstruction result on K-RuO₂ (Figure S17-19) also indicates that K-RuO₂ does not possess a nanoreactor structure like in MD-RuO₂-BN. Therefore, the liquid environment and structure-oriented effects generated by eutectic

molten salts induce the formation of ultrafine, uniform and continuous RuO_2 nanomonomers, and periodically regular porous structures with 3D interconnections (Figure S20), which are considered important conditions for formation of bicontinuous nanoreactors.”

Figure R1. (a, b) STEM images of MD-RuO₂-BN. (c, d) STEM images of C-RuO₂.

Figure R2. (a-c) STEM images of K-RuO₂ (RuO₂ product made in KCl alone).

Figure R3. 70 STEM-HAADF images of K-RuO₂ for tomography reconstruction.

Figure R4. Reconstructed model of K-RuO₂.

Figure R5. (a-c) Corresponding front, top and right view of reconstructed K-RuO₂. (d) Extracted cubic sub volume from the labeled red dash line area in figure (a-c). (e) The right view of sub volume. (f) Representative ortho slices marked by black dash line in figure (e). (g) Volumes from segmentation by contrast corresponding to RuO₂ (blue) and void (black), respectively.

Q2. The synthetic route is very interesting, and the outcome looks promising. However, with only a single sample prepared, the results are not yet convincing. The benchmark used is C-RuO₂. If the authors are trying to prove that using a eutectic solution is a superior method, they must demonstrate that a RuO₂ product made in KCl alone is of inferior quality to MD-RuO₂-BN. I am concerned that the authors have built a massive theory and explanation based on just a SINGLE sample. Please run a blank experiment using KCl instead of KCl-LiCl as a control. Please critically evaluate your data and be honest about the limitations before overselling your paper based on a single sample.

More replicates (which are prepared under similar conditions) are needed to substantiate the claims.

Reply: Thanks for your professional comments. As suggested, we have carried out a blank experiment using KCl instead of KCl-LiCl eutectic system as a control under the same conditions (K-RuO₂). The XRD pattern (**Figure R6a**) shows that as-prepared K-RuO₂ has the same crystal structure and phase as rutile RuO₂ (JCPDS 70-2662). By focusing on its individual nanoparticles, the high-resolution atomic images show the lattice fringes corresponding to the (110) plane (**Figure R6b-c**) of RuO₂, further demonstrating this.

Next, we have evaluated the OER performance of K-RuO₂ under the same testing conditions and compared it with MD-RuO₂-BN and C-RuO₂. As shown in **Figure R7a**, polarization curves demonstrate the overpotential of K-RuO₂ (245 @10 mA·cm⁻²) is lower than C-RuO₂ (305 @10 mA·cm⁻²), but still significantly higher than MD-RuO₂-BN (196 @10 mA·cm⁻²), proving that using a eutectic solution is a superior method. Moreover, compared with K-RuO₂ and C-RuO₂, we can find that MD-RuO₂-BN possesses the lower Tafel slope (**Figure R7b**), smaller charge-transfer resistance (**Figure R7c**) and higher electrochemically active surface area (**Figure R8**) during OER processes. By combining with the microscopic-electronic structure characterization above, it further confirms a wide range of active interfaces through particle size refinement and full penetration of electrolyte for the unique bicontinuous nanoreactor in MD-RuO₂-BN. Furthermore, for MD-RuO₂-BN, the constant current chronopotentiometry (**Figure R7d**) exhibits the increased OER potential by only 0.029 V after 24 h continuous operation, indicating its significantly better OER stability than that of K-RuO₂ and C-RuO₂.

The above construction and performance testing of K-RuO₂ further highlight the advantages of KCl-LiCl eutectic system. The liquid environment triggered by it is the key to forming a dual continuous nanoreactor and inducing excellent catalytic performance.

We have made a corresponding revision in the manuscript, and the relevant figures have been added in the revised manuscript and supporting information.

The revision is shown as below:

(Page 15) “As shown in **Figure 4a**, polarization curves demonstrate a remarkably low overpotential of MD-RuO₂-BN (196 @10 mA·cm⁻²) compared with that of K-RuO₂ (245 @10 mA·cm⁻²) and C-RuO₂ (305 @10 mA·cm⁻²), indicating that the structural design and defect engineering of RuO₂ nanoreactors can achieve higher activity. Moreover, we can find that MD-RuO₂-BN possesses the lower Tafel slope (**Figure 4b**), smaller charge-transfer resistance (**Figure 4c**) and higher electrochemically active surface area (Figure S24) during OER processes.”

(Page 15-16) “Furthermore, the constant current chronopotentiometry (**Figure 4d**) exhibits the increased OER potential by only 0.029 V after 24 h continuous operation (degradation rate of 1.2 mV h⁻¹), indicating the higher OER stability of MD-RuO₂-BN than that of K-RuO₂ (5.3 mV h⁻¹) and C-RuO₂ (rapid degradation).”

Figure R6. (a) XRD pattern of K-RuO₂. (b) STEM image of K-RuO₂. (c) Corresponding high-resolution atomic image from the area indicated by the yellow box in figure (b).

Figure R7. (a) Polarization curves, (b) Tafel curves and (c) Nyquist plots of MD-RuO₂-BN, C-RuO₂, and K-RuO₂ for OER. (d) Constant current stability testing for OER.

Figure R8. Cyclic voltammograms of (a) MD-RuO₂-BN, (c) C-RuO₂ and (e) K-RuO₂ in the region of (0.70) - (0.80) V versus SCE at different scan rates. Corresponding

linear relationships between capacitive current and scan rate of (b) MD-RuO₂-BN, (d) C-RuO₂ and (f) K-RuO₂.

Q3. Looking at the results in Figure 3, the difference between MD-RuO₂-BN and C-RuO₂ seems negligible. However, as mentioned in point 2, these two products are very different in preparation. Since the authors claim a unique ability of the eutectic solution to create a superior catalyst, it is important to compare to a blank sample prepared using just KCl. Despite the samples C-RuO₂ and MD-RuO₂-BN are different, unlike the authors, I have doubts there is an appreciable change in Ru oxidation state, and if there was, the highly oxidative conditions at the anode would likely oxidize Ru to a high state anyway. Can the authors explain their conclusions in the context of actual processes likely to occur at the anode? If not, it is perfectly acceptable to avoid confusing the readers (given the general readership of Nature Communications), with terminology like "unique bi-continuous nanoreactors" when the evidence is lacking.

Reply: We are very appreciative of your thoughtful comments. Blank samples prepared using just KCl (K-RuO₂) have been characterized accordingly and compared with MD-RuO₂-BN and C-RuO₂. First, as discussed in question 1 (**Q1**), due to the lack of a liquid environment, the grain size of K-RuO₂ (**Figure R9b**) is larger and uneven compared with MD-RuO₂-BN (**Figure R9a**). And like C-RuO₂ (**Figure R9c**), it also does not present a special nanoreactor structure. Further atomic phase observation (**Figure R10**) reveals that K-RuO₂ crystals do not have intra granular defects found in MD-RuO₂-BN. Therefore, K-RuO₂ is more likely to maintain consistency in electronic structures with C-RuO₂. Indeed, the results of XPS and EPR also confirm this. As shown in **Figure R11a**, C-RuO₂ and K-RuO₂ have almost identical Ru 3p spectra, but two peaks of Ru 3p_{3/2} and Ru 3p_{1/2} for MD-RuO₂-BN shift about 0.3 eV to lower binding energy relative to C-RuO₂ and K-RuO₂, implying the formation of more low charge Ru ions and oxygen vacancy defects (V_O). In O 1s spectra (**Figure R11b**), the peak proportion attributable to V_O of MD-RuO₂-BN, C-RuO₂ and K-RuO₂ is 46.2%, 29.7%, and 31.8%, respectively, indicating a significant increase in V_O concentration in the prepared MD-RuO₂-BN. Furthermore, MD-RuO₂-BN shows a stronger EPR signal at g = 2.003

(**Figure R11c**), also proves that MD-RuO₂-BN contained more V_O than C-RuO₂ and K-RuO₂.

In fact, to explain our conclusions in the context of actual processes likely to occur at the anode, the in-situ Raman spectra were performed in original manuscript. As shown in **Figure R12a** and **R12b**, the three major Raman features of rutile RuO₂, namely the E_g (518 cm⁻¹), A_{1g} (642 cm⁻¹) and B_{2g} (702 cm⁻¹) vibration modes can be observed both on MD-RuO₂-BN and C-RuO₂ in ordinary 0.5 M H₂SO₄. Besides, two Raman bands at about 430 and 588 cm⁻¹ assign to Ru⁴⁺-O bonds and Ru³⁺-O bonds, respectively (*Nat. Commun.* **14**, 2517 (2023)). Significantly, when further normalizing the intensity of the band at 588 cm⁻¹ and 430 cm⁻¹ (**Figure R12c**), a higher intensity can be found for MD-RuO₂-BN (about 1.0) in comparison with C-RuO₂ (about 0.4), thereby demonstrating that MD-RuO₂-BN possesses more Ru³⁺ species on the surface, further supporting the higher content of low-valent Ru species on the MD-RuO₂-BN surface caused by multiscale defects.

In addition, we have made a corresponding revision in the manuscript. And the relevant figures have been added in the revised manuscript and supporting information. The revision is shown as below:

(Page 11-12) “(**Figure 3e**) shows that C-RuO₂ and K-RuO₂ have almost identical Ru 3p spectra, but two peaks of Ru 3p_{3/2} and Ru 3p_{1/2} for MD-RuO₂-BN shift about 0.3 eV to lower binding energy relative to C-RuO₂ and K-RuO₂, implying the formation of more low charge Ru ions and oxygen vacancy defects (V_O). In O 1s spectra (**Figure 3f**), the peak proportion attributable to V_O of MD-RuO₂-BN, C-RuO₂ and K-RuO₂ is 46.2%, 29.7%, and 31.8%, respectively, indicating an obvious increase of V_O concentration in MD-RuO₂-BN.^{15, 18} Besides, the binding energy position of the Ru-O characteristic peak for MD-RuO₂-BN also shifts by about 0.1 eV relative to C-RuO₂ and K-RuO₂, further suggesting a redistribution of charges. Moreover, MD-RuO₂-BN shows a stronger electron paramagnetic resonance (EPR) signal at g = 2.003, also proving that MD-RuO₂-BN contains more V_O than C-RuO₂ and K-RuO₂ (**Figure 3g**).”

Figure R9. STEM images of (a) MD-RuO₂-BN, (b) K-RuO₂, and (c) C-RuO₂.

Figure R10. High-resolution atomic images of (a) MD-RuO₂-BN, and (b) K-RuO₂.

Figure R11. XPS spectra of Ru 3p (a) and O 1s (b) for MD-RuO₂-BN, C-RuO₂ and K-RuO₂. (c) EPR spectra of MD-RuO₂-BN, C-RuO₂ and K-RuO₂.

Figure R12. (a, b) Raman spectra for MD-RuO₂-BN and C-RuO₂ in ordinary 0.5 M H₂SO₄. (c) Normalized intensity of Raman band at 588 cm⁻¹ to that at 430 cm⁻¹ on the catalysts as a function of applied potential.

Q4. What was the dissolution rate of the control C-RuO₂ in H₂SO₄ according to ICP? What would be the dissolution rate of the control RuO₂ prepared in KCl?

Reply: Thanks for your valuable comments. The concentration of Ru ions in the electrolyte was measured using inductively coupled plasma-optical emission spectroscopy (ICP-OES). The percentage of Ru dissolved from C-RuO₂ and K-RuO₂ during the OER is 5.6 % and 4.5 %, respectively. For MD-RuO₂-BN, the dissolution rate of Ru is only 2.9 %. Hence, the bicontinuous nanoreactor structure and the optimized Ru coordination environment are considered to be the main reasons for the stability improvement, because the rapid release of gas and the weakening of Ru-O interaction effectively prohibit the formation and dissolution of high valence Ru ions.

Q5. It is fine to carry out tests at negligibly low current densities of 10 mA/cm² in sulfuric acid as it allows for the comparison with literature. However, authors provided no details (also not in the SI) about the testing conditions. What was the size of glassy carbon electrode used, for example? Please provide comprehensive information and also run the experiment on a blank prepared in KCl.

Reply: Thanks for your kind reminder. We have added all three-electrode testing details to the SI as below:

“All electrochemical measurements were performed in a conventional three-electrode system at room temperature using a CHI 660E electrochemical analyzer (CHI Instruments, Shanghai, China). The acidic (0.5 M H₂SO₄) electrochemical measurements were performed using a saturated calomel electrode (SCE) as the reference electrode, a graphite plate as the counter electrode, and a glassy carbon electrode with a diameter of 3 mm as the working electrode. The catalyst ink was prepared by dispersing 5 mg as-prepared sample into a mixture (900 μL isopropyl alcohol, 100 μL water and 20 μL 5% Nafion solution) and ultrasonic dispersion for 30 min. For comparison, 5 mg commercial catalyst powder (RuO₂ or 20 wt% Pt/C) was evenly dispersed into the same mixture. Polarization data were obtained at a scan rate of 5 mV s⁻¹. In this work, all potentials measured against SCE were converted to the reversible hydrogen electrode (RHE) scale using: E (potential, versus RHE) = E (versus SCE) + 0.241 V + 0.0591 × pH. All polarization curves were iR-corrected. The electrochemical impedance spectroscopy (EIS) was conducted at the corresponding potentials of 10 mA cm⁻² from LSV curves, with the frequency range of 0.01 Hz to 100 kHz with AC amplitude of 10 mV. The electrochemical double layer capacitance (C_{dl}) was determined with typical cyclic voltammetry (CV) measurements at various scan rates (20, 40, 60, 80 and 100 mV s⁻¹) in nonreactive region. The durability was evaluated by accelerated degradation measurements and constant current chronopotentiometry. The obtained electrocatalyst and Pt/C were used as anode and cathode in a two-electrode configuration for overall water splitting. And the generated H₂ and O₂ gases during overall water splitting were quantitatively collected by the water drainage method for evaluating Faraday efficiency.”

Besides, as shown in **Figure R7** and **R8**, we have presented the test results of K-RuO₂ in the three-electrode system and compared them with MD-RuO₂-BN and C-RuO₂, and the specific analysis and discussion have been supplemented in the reply to question 2 (**Q2**).

Q6. My primary concern is the performance in the flow-cell PEMWE electrolyser (Fig. 4g). The MD-RuO₂-BN reaches 1A/cm² at a relatively low potential (below 2V), as expected given its high surface area and small RuO₂ nanoparticles. However, the high potential seemingly required for C-RuO₂ is unexpected. We routinely reach 1A/cm² below 2V by running a LSV with commercial RuO₂ in my lab (including those immobilised on Ti GDL by spraying). The LSV curve for RuO₂ does not seem correct - there were likely serious issues in preparation. There could be issues with the electrode preparation and spraying process, which the authors describe insufficiently. Please provide comprehensive details on the preparation methods. The authors should re-run tests with C- RuO₂ and RuO₂ in KCl for comparison. Please update both methods and preparation sections both in the main text and SI. SEM imaging of the RuO₂ immobilized on Ti GDL would also be more useful than the 3-electrode system work. Overall, more information is needed on the electrode preparation and benchmarking experiments. As mentioned below the stability of MD-RuO₂-BN is important achievement but given the issues with benchmark and unusual LSV for C-RuO₂ more experiments (and description of the experiments) are required.

Reply: We're very appreciative of your constructive comments on the flow-cell PEMWEs, which has significantly helped us to improve the quality of our work. As suggested, to improve the performance of PEMWEs, we have optimized the preparation of catalyst-coated membrane (CCM) and the assembly of the electrolyzer, as well as upgrading the testing system. Herein, we first provide comprehensive details on the preparation methods and add these details to the SI as below:

“OER activities of MD-RuO₂-BN, K-RuO₂ and C-RuO₂ in membrane-based systems were evaluated in PEMWEs consisting of CCM, porous transport layers (PTL) and bipolar plates. The cathode catalyst ink was prepared by mixing 35 mg Pt/C (40 wt%) powder, 300 mg of Nafion solution (5 wt%), 2 mL of DI water and 8 mL of isopropanol and then sonicated for 60 min in an ice bath. The anode catalyst ink was prepared by mixing 40 mg of RuO₂ powder, 200 mg of Nafion solution (5 wt%), 1 mL of DI water and 4 mL of isopropanol and then sonicated for 60 min in an ice bath. The mass ratio of Nafion in the cathode and anode catalyst layer was 30 wt% and 20 wt%, respectively.

To prepare the CCM with Nafion 115 membrane (127 μm) as an electrolyte, the anode and cathode catalysts were first sprayed onto sheets of polytetrafluoroethylene (PTFE). After that, the PTFE-supported Pt/C, Nafion 115, and PTFE-supported anode catalysts were hot pressed together at 130 $^{\circ}\text{C}$ for 10 min under a pressure of 20 MPa. After cooling, the PTFE on the surface were carefully peeled off to get the CCM (**Figure R13**). The anode and cathode catalysts loading were controlled to be 2 mg cm^{-2} and 1 mg cm^{-2} after loading optimization. The CCM prepared was preserved in distilled water for further measurements. To construct the PEM electrolyzers for performance evaluation, the titanium felt with a thickness of 6 mm were used as the PTL in both the anode and cathode. The assembly pressure of the fixture is set to 8 N m. The active area of the electrode was measured to be 4 cm^2 . The PEM electrolyzers were operated at 80 $^{\circ}\text{C}$ with distilled water as reactant under a flow rate of 30 mL min^{-1} . Before the polarization test, the cell was activated for 1 h at 1 A cm^2 . V-I curves were measured to evaluate the effects of anode catalysts on PEMWEs performances. Long-term PEMWEs performances were evaluated by measuring the current at constant cell potential. All the data of PEMWEs were not iR corrected and displayed as raw data.”

Compared to the previous preparation process, we have optimized the Nafion content in the catalyst paste, improved the coating process of the catalyst, and updated the Nafion membrane that may have deteriorated. In addition, we have optimized the contact between CCM and PTL during the assembly process of PEMWEs. Finally, we adopted a better working platform (CAMRY,30 A range).

As shown in **Figure R14**, the whole testing system consists of a GAMRY electrochemical workstation, electrolyzer, heating rod, peristaltic pump and water tank. Furthermore, to ensure sufficient and uniform contact between the catalytic layers and PTL, we have simulated the distribution of surface stress using pressure-sensitive paper in advance. As shown in **Figure R15**, under an assembly pressure of 8 N m, the surface stress distribution nephogram of PTL and pressure-sensitive paper is very uniform, indicating that the catalytic layers had good contact with PTL under such pressure,

conducive to achieving excellent performance.

After improving the assembly process and test condition, we have retested and compared the performance of the catalysts in the flow-cell PEMWEs. The polarization curves of MD-RuO₂-BN-PEM-Pt/C, K-RuO₂-PEM-Pt/C and C-RuO₂-PEM-Pt/C show that the water electrolysis activity of PEMWEs can be greatly enhanced by adopting MD-RuO₂-BN as anode catalyst (**Figure R16**). Specifically, to reach a current density of 1 A cm⁻² for water electrolysis, MD-RuO₂-BN-PEM-Pt/C only requires a cell voltage of 1.64 V, superior to that obtained with K-RuO₂-PEM-Pt/C (1.72 V@1 A cm⁻²) and C-RuO₂-PEM-Pt/C (1.85 V@1 A cm⁻²). Besides, stability testing results will be presented in the following question (**Q7**).

Moreover, we have also provided cross-sectional and planar SEM images of CCM with MD-RuO₂-BN as anode catalysts (**Figure R17**). It can be observed that the MD-RuO₂-BN catalytic layer (CL) is uniformly coated on the Nafion membrane and the CL thickness was estimated to be 20 μm.

In addition, we have updated all PEMWEs test results. Other relevant figures have also been added to revised manuscript and supporting information. The revision is shown as below:

(Page 16-17) “Therefore, a PEMWE single cell consisting of catalyst (MD-RuO₂-BN at anode and 40% Pt/C at cathode) coated membrane (Nafion® 115) (CCM), porous transport layers (PTL) and bipolar plates was finally installed (**Figure 4e, f**) and tested (Figure S33). Surface stress distribution nephogram of PTL and pressure-sensitive paper suggests that the good contact of the catalytic layers with PTL under an assembly pressure of 8 N m (Figure S34). Besides, the cross-sectional and planar SEM images of the CCM with MD-RuO₂-BN (Figure S35) show that the catalytic layer is uniformly coated on the membrane and the thickness of catalytic layer was estimated to be 20 μm. The polarization curves (**Figure 4g**) of PEMWEs shows that, to reach a current density of 1 A cm⁻² for water electrolysis, MD-RuO₂-BN-PEM-Pt/C with MD-RuO₂-BN as

anode catalyst only requires a cell voltage of 1.64 V, superior to that obtained with K-RuO₂-PEM-Pt/C (1.72 V@1 A cm⁻²) and C-RuO₂-PEM-Pt/C (1.85 V@1 A cm⁻²).”

Figure R13. Photographs of the CCM, which consists of anode catalytic layer (MD-RuO₂-BN), Nafion 115 membrane and cathode catalytic layer (Pt/C).

Figure R14. Photographs of the PEMWE testing system.

Figure R15. (a) Photograph of pressure-sensitive paper. (b) Surface stress distribution nephogram of PTL and pressure-sensitive paper.

Figure R16. Polarization curves of the PEM electrolyzer measured at 80 °C utilizing the as-synthesized MD-RuO₂-BN, K-RuO₂ or C-RuO₂ as an anode and 40% Pt/C as a cathode.

Figure R17. (a) Planar and (b) cross-sectional SEM images of MD-RuO₂-BN coated membrane.

Q7. Figure 4h shows MD-RuO₂-BN has good stability at 0.2 A/cm², but the applied potential is not reported. Please rerun the test at 2 V (which should yield ~1 A/cm² based on the LSV), as this is a realistic potential used in good performing electrolyzers. Again, it is important to run a similar stability test on RuO₂ prepared in KCl alone for comparison. Again, the fading of C-RuO₂ at such a quick rate in just 0.2 A/cm² makes me concerned about preparation of the electrodes. More work is needed here as this seems to be the key to application of MD-RuO₂-BN.

Reply: Thanks for your professional comments. Based on your suggestion, we have retested the stability of the improved electrolyzers. The test condition is shown in **Q6**. Specifically, we have selected 1.65 V, 1.72 V, and 1.85 V as the constant voltages for stability tests of MD-RuO₂-BN-PEM-Pt/C, K-RuO₂-PEM-Pt/C and C-RuO₂-PEM-Pt/C, respectively (which should yield ~1 A/cm² based on the LSV). Time-dependent current density curves (**Figure R18**) reveal that our MD-RuO₂-BN-based electrolyzer well-maintains water electrolysis activity for 50 h, while both K-RuO₂ and C-RuO₂ experiences significant decline within 25 h of operation, especially C-RuO₂. Therefore, MD-RuO₂-BN is superior to K-RuO₂ and C-RuO₂ in practical applications. This is consistent with their performance in the three-electrode test, further proving that our design has improved the stability of RuO₂.

In addition, we have made corresponding modifications to the manuscript and updated the testing results and figure in the revised manuscript. The revision is shown as below:

(Page 17) “Besides, we selected 1.65, 1.72, and 1.85 V as the constant voltage for stability tests of MD-RuO₂-BN-PEM-Pt/C, K-RuO₂-PEM-Pt/C and C-RuO₂-PEM-Pt/C (which should yield ~1 A/cm² based on the LSV), respectively. Time-dependent current density curves (**Figure 4h**) reveal that our MD-RuO₂-BN-based electrolyzer well-maintained water electrolysis activity for 50 h, while both K-RuO₂ and C-RuO₂ experienced significant decline within only 25 h of operation, especially C-RuO₂. This is consistent with their performance in the three-electrode test, further proving that our designed catalyst can improve the stability of RuO₂.”

Figure R18. Time-dependent current density curves of MD-RuO₂-BN, K-RuO₂ and C-RuO₂ catalysts in the PEMWE using commercial Pt/C as the cathode.

Q8. The isotope exchange results are quite interesting, but as mentioned above, have limited relevance from an applications perspective since the 3-electrode testing was done at incredibly low current densities of 10 mA/cm². This testing provides little insight into how the material would perform under actual PEM electrolysis conditions. It would be more useful to provide photos and SEM images of the electrodes after PEMWE testing, rather than spending so much effort on highly complicated 3-electrode experiments at non-representative current densities. While academically interesting, the priority should be characterizing performance at industrially-relevant currents in a PEMWE cell, rather than low current density 3-electrode testing.

Reply: Thanks for your constructive insights and comments, which have prompted us to optimize the performance of PEMWEs. Moreover, as suggested, we have provided photos (**Figure R19**) and SEM images (**Figure R20**) of the electrodes after PEMWE testing. **Figure R19** shows that there is no detachment of the MD-RuO₂-BN catalytic layer on anode after operation at 1 A cm⁻² for 50 h. And the cross-sectional and planar morphology of the MEA further demonstrate the catalytic layer and membrane structure were well preserved (**Figure R20**).

In addition, we have made a corresponding revision in the manuscript. And the relevant figures have been added in the supporting information. The revision is shown as below: (Page 17) “Moreover, Figure S36 shows that there is no detachment of the MD-RuO₂-BN catalytic layer on anode after operation at 1 A cm⁻² for 50 h. And the cross-sectional and planar morphology of the MEA further demonstrate the catalytic layer and membrane structure are well preserved (Figure S37).”

Figure R19. Photos of the electrodes before (a) and after (b) PEMWE testing.

Figure R20. (a) Planar and (b) cross-sectional SEM images of MD-RuO₂-BN coated membrane before PEMWE testing. (c) Planar and (d) cross-sectional SEM images of MD-RuO₂-BN coated membrane after PEMWE testing.

Response to Reviewer #2

The author used liquid molten salt to trigger the dual modulation of Ru electronic characteristics and local microenvironments, and successfully synthesized a heterogeneous multiscale defective RuO₂ “bicontinuous nanoreactors” catalyst. The synthesized MD-RuO₂-BN exhibited a better alkaline oxygen evolution performance than the commercial RuO₂ (C-RuO₂). Specifically, RuO₂-BN delivered an overpotential of 196 mV (vs RHE) at a current density of 10 mA cm⁻² in electrochemical tests and 2.02 V at a current density of 1 A cm⁻² in PEMWE devices. We appreciate the authors’ efforts in this research. However, there are many contradictions in the manuscript and the lack of innovations. Some similar works have been published previously (Nat. Commun. 13, 5716 (2022); Nat. Commun. 14, 1412 (2023)). The above issues prevent us from recommending the manuscript for publication in Nature Communications.

Reply: First of all, we are very grateful for your comments on our work. We emphasize that we really respect your point of view and have made detailed point-by-point responses and revisions based on your feedback, which has greatly contributed to the improvement of this article.

For the novelty, we have carefully read the two articles you recommended, and benefited from the high quality of the presentation. However, such two articles and our article have significant differences in catalyst design, optimization strategies, and core viewpoints. As you can see, the first article (*Nat. Commun.* 13, 5716 (2022)) mainly focused on breaking the stability and activity limits of RuO₂ in neutral and alkaline environments by constructing a RuO₂/CoO_x interface (**Figure R21-a**). Specifically, the sacrificial oxidization of CoO_x and the electron interaction among the face-to-face Ru-O-Co interfacial atoms enhanced the stability, while the Ru/Co dual-atom site exposed around the interface was responsible for the improved activity. And the second article (*Nat. Commun.* 14, 1412 (2023)) reported a synergistic strategy of Rh doping and surface oxygen vacancies to precisely regulate unconventional OER reaction path via the Ru-O-Rh active sites of Rh-RuO₂, simultaneously boosting intrinsic activity and stability (**Figure R21-b**). Obviously, these are very different from our design ideas in

this work: we pay more attention to the structural and intrinsic activity improvements of RuO₂ without introducing foreign elements, heterostructures or carriers. Specifically, a unique bicontinuous nanoreactor composed of numerous multiscale defected RuO₂ nanocrystals and internal pores is conceived to efficiently and stably drive acid water oxidation, demonstrating a promising application in PEMWEs and integrated regenerative fuel cells as hydrogen-water circulating power supply systems.

Figure R21. (a) Schematic diagram of the interfacial structure of RuO₂/CoO_x, (*Nat. Commun.* 13, 5716 (2022)). (b) Schematic diagram of Rh-RuO₂/G nanosheets, (*Nat. Commun.* 14, 1412 (2023)).

In order to better obtain the importance of our work, we have highlighted the scientific discovery, novelty and significance of our work as follows:

(1) In the structural aspects, a multiscale defected RuO₂ bicontinuous nanoreactor (MD-RuO₂-BN) is conceived and confirmed by advanced three-dimensional tomograph reconstruction technology. This unique structure not only provides abundant active sites and reaction regions for the catalytic process, but also enhances electron and mass transfer through a cavity confinement effect.

(2) In the synthesis aspects, we pioneer the use of the liquid molten salt (KCl-LiCl) to trigger the dual modulation of Ru electronic characteristics and local microenvironments. For comparison, the grain size of K-RuO₂ (synthesized under the same conditions in the presence of only KCl) is large, uneven and disorderly arranged, demonstrating the liquid molten salt is an important factor in the formation of nanoreactors.

(3) In the atomic and local electronic structure aspects, our investigation indicates that there are rich Ru vacancy defects (V_{Ru}), oxygen vacancy defects (V_O), intra/inter granular boundaries and other multi-scale defects from points to surfaces in the ultrafine RuO_2 nanocrystals. These defects endow RuO_2 generous low coordination Ru atoms and weakened Ru-O interaction, greatly inhibiting the oxidation of lattice oxygen and dissolution of high-valence Ru.

(4) In the electrochemical performance aspects, as expected, the MD- RuO_2 -BN achieves unparallel OER activity (196 mV @ 10 mA cm⁻²) and an ultralow degradation rate of 1.2 mV h⁻¹ in acidic media. We also prove a high-performance PEMWE with MD- RuO_2 -BN as anode electrocatalyst delivers a cell voltage of 1.64 V at 1 A cm⁻². And the successful demonstration of the integrated hydrogen-water circulating power supply system further provides more opportunities for practical applications.

(5) In the structure-effect relationship aspects, both density functional theory (DFT) calculations and in-situ technology further unveil the electronic structure of MD- RuO_2 -BN and the mechanism of water oxidation process. The synergistic effect of the multiscale defects and the protected active Ru sites contributes to improvement of activity and stability.

(6) Profoundly, this work provides a new insight into improving catalytic performance of the Ir-free-based OER catalyst, and will stimulate the development of PEMWEs for large-scale green H₂ generation. More importantly, the optimization strategy of electron- and micro-structure synchronization induced by the construction of a unique bicontinuous nanoreactor will benefit many researchers in nanoresearch field and beyond.

Besides, we have taken all the professional and constructive comments you raised (attached below). And once again, we appreciate your efforts in the review process. Based on resolving these issues, and in particular after we have conducted some further experimental work, we hope that the revised version can receive your favorable consideration.

Q1. The synthesis mechanism of the “bicontinuous nanoreactors” needed to be elaborated in detail.

Reply: We are very appreciative of your thoughtful comments. Here, the liquid molten salt formed by KCl-LiCl eutectic at high temperatures is an important medium for the formation of RuO₂ nanoreactors. Specifically, as shown in **Figure R22**, when the precursor and eutectic salt form a uniform mixture, in the subsequent pyrolysis process, the molten state of the eutectic salt can create an ionic liquid-confined space, providing a uniform growth environment for RuO₂ crystal to avoid agglomeration and the formation of large particles. Meanwhile, liquid molten salt also acts as a structural guide to adjust the structural characteristics of the generated solids, that is, to leave a large number of uniformly distributed and continuous holes for the RuO₂ crystal during the subsequent cooling separation process. Therefore, under the triggering of liquid molten salt, a bicontinuous nanoreactors composed of RuO₂ ultrafine monomers and internal pores was ultimately formed.

Furthermore, we have added the nanostructure characterization (**Figure R23**) of RuO₂ synthesized under the same conditions in the presence of only KCl (K-RuO₂). Here, due to the much higher melting point of KCl (770 °C) than the synthesis temperature (500 °C), the reaction system is unable to produce a liquid reaction medium. It can be found that the average grain size of K-RuO₂ (~10 nm) is significantly larger than MD-RuO₂-BN (~3 nm). Importantly, due to the lack of a liquid medium to provide a uniform growth environment, the grain size of K-RuO₂ is uneven and disorderly arranged. 70 STEM-HAADF images for 3D tomography reconstruction on the obtained K-RuO₂ were collected by a 1-2° interval (**Figure R24**). The resulting reconstructed structural unit is displayed in **Figure R25**. The sub volume extraction and segmentation (**Figure R26**) indicate that K-RuO₂ does not possess a nanoreactor structure like in MD-RuO₂-BN, further demonstrated the liquid molten salt is an important factor in the formation of nanoreactor.

In addition, we have made a corresponding revision in the manuscript. And the relevant figure has been added in the supporting information.

Figure R22. Schematic illustration of the synthesis of the RuO₂ bicontinuous nanoreactors.

Figure R23. (a-c) STEM images of K-RuO₂. (d) Corresponding high-resolution atomic image from the area indicated by the yellow box in figure (c). (e-h) STEM mapping and the corresponding elemental distribution of K-RuO₂.

Figure R24. 70 STEM-HAADF images of K-RuO₂ for tomography reconstruction.

Figure R25. Reconstructed model of K-RuO₂.

Figure R26. (a-c) Corresponding front, top and right view of reconstructed K-RuO₂. (d) Extracted cubic sub volume from the labeled red dash line area in figure (a-c). (e) The right view of sub volume. (f) Representative ortho slices marked by black dash line in figure (e). (g) Volumes from segmentation by contrast corresponding to RuO₂ (blue) and void (black), respectively.

Q2. Why named the MD-RuO₂-BN “bicontinuous nanoreactors”? In fact, many kinds of RuO₂ have a pore structure, could they all be categorized as nanoreactors? (Chinese Journal of Chemical Engineering 55 (2023) 93-10).

Reply: Thanks for your professional comments. The structural characterization indicates that the MD-RuO₂-BN catalyst we prepared is composed of numerous ultrafine RuO₂ monomers with a particle size of approximately 3nm (**Figure R27**). And advanced three-dimensional tomograph reconstruction technology has been used to prove that the RuO₂ monomers and internal pore structures are continuous, rather than

enclosed or isolated (**Figure R28**). Therefore, we named the catalyst with the unique structure as the bicontinuous nanoreactors.

Of course, not all materials with pore structures can be called nanoreactors. First of all, nanoreactor refers to a mesoscopic environment where chemical reactions are limited by the nanoscale space (*Angew. Chem. Int. Ed.* **2022**, *61*, e202204371; *Adv. Funct. Mater.* **2022**, *32*, 2205569). Therefore, ultrafine, uniform, and continuous monomers are important conditions for forming a nanoreactor. Moreover, the nanoreactor should possess a periodically regular porous structure with three-dimensional interconnections. If the particles are too large and uneven or the pore structure is dispersed and not connected with each other, it is difficult to form a cavity with enrichment effect, thus slowing down mass transfer and reaction kinetics. For example, the RuO₂@Ru/HCs shown in the article you recommended (*Chin. J. Chem. Eng.* **55** (2023) 93-10) is a typical hollow core-shell structure, but it cannot be called a nanoreactor due to the presence of layered pore structures only distributed on the catalyst surface and the obvious aggregation of Ru particles. And the obtained K-RuO₂ (**Figure R23-26**) also does not possess a nanoreactor structure like in MD-RuO₂-BN.

In addition, we have made a corresponding revision in the manuscript. And the relevant figure has been added in the supporting information.

Figure R27. (a) STEM image of MD-RuO₂-BN. (b) Gaussian fitted size distribution of RuO₂ nanoparticles.

Figure R28. (a, b) Representative STEM-HAADF image and reconstructed MD-RuO₂-BN at front view. (c) Extracted cubic sub volume from the labeled yellow dash line area in figure (b). (d) The right view of sub volume. (e) Representative ortho slices marked by black dash line in figure (d).

Q3. In the XRD patterns of MD-RuO₂-BN and C-RuO₂ (Figure 1e), the diffraction peak of MD-RuO₂ was weaker than that of C-RuO₂ obviously.

Reply: Thanks for your kind reminder. We are very sorry for the error in the description of **Figure 1e** in the original manuscript. As you can see, the diffraction peak of MD-RuO₂-BN is significantly wider and weaker than that of C-RuO₂, implying formation of small-sized nanoparticles (*Adv. Sci.* 9, 2200010 (2022); *Small* 16, 2002124 (2020)). Furthermore, this conclusion is further supported by comparing the microstructure of MD-RuO₂-BN (**Figure R29a, b**) and C-RuO₂ (**Figure R29c, d**) observed by TEM. Apparently, the RuO₂ unit in MD-RuO₂-BN is smaller and uniform, while the C-RuO₂

possesses larger and uneven grain size, with a disordered spatial structure and obvious agglomeration.

In addition, we have made a corresponding revision to the description of **Figure 1e** in the revised manuscript:

“Notably, the diffraction peak of MD-RuO₂ becomes wider and weaker than that of commercial RuO₂ (C-RuO₂), implying formation of small-sized nanoparticles, consistent with the above TEM observation results.”

Thank you again for carefully reviewing and identifying the issues for us.

Figure R29. (a, b) STEM images of MD-RuO₂-BN. (c, d) STEM images of C-RuO₂.

Q4. In the Ru 3p XPS spectra of MD-RuO₂-BN and C-RuO₂ (Figure 3e), two peaks assigned to Ru 3p of C-RuO₂ were located at higher energy than that of MD-RuO₂-BN, indicating a higher Ru valence state in C-RuO₂ than MD-RuO₂-BN. However, the author's description here was “C-RuO₂ shift about 0.3 eV to lower binding energy relative to the obtained MD-RuO₂-BN”. In addition, previous studies revealed that Ru species with a higher oxidation state could enhance their OER activity. Why do the lower valence states of Ru species have better OER performance in this manuscript? (Nat. Mater. 22, 100-108 (2023)).

Reply: Thanks for your constructive comments. After careful examination, we confirm that there is no problem with the data in **Figure 3e**. We apologize for reversing the positions of C-RuO₂ and MD-RuO₂-BN in the description of **Figure 3e**. We have made corresponding modifications in the revised manuscript:

“The X-ray photoelectron spectroscopy (XPS)(**Figure 3e**) shows that C-RuO₂ and K-RuO₂ have almost identical Ru 3p spectra, but two peaks of Ru 3p_{3/2} and Ru 3p_{1/2} for MD-RuO₂-BN shift about 0.3 eV to lower binding energy relative to C-RuO₂ and K-RuO₂, implying the formation of more low charge Ru ions and oxygen vacancy defects (V_o).”

Furthermore, the shift of the absorption edge in the X-ray absorption near edge structure (XANES) of Ru K-edge further supports this conclusion (**Figure 3h**).

As for whether Ru with a higher oxidation state or a lower oxidation state can enhance the OER activity of the catalyst, we believe that we cannot ignore the specific catalyst design (electronic structure, coordination environment, etc.) while only consider the effect of valence states on performance. For example, in the article you recommended (Nat. Mater. 22, 100-108 (2023)), Ni doped RuO₂ indeed increased the oxidation state of Ru and enhanced the activity of OER, but more importantly, it stabilized the lattice of RuO₂, greatly improving the stability of catalysts in acidic media. However, in many reported articles, RuO₂ with lower oxidation states also exhibits excellent OER activity. For instance, among the designed catalysts such as Li_xRuO₂ (Nat. Commun., 2022, 13,

3784), Nb_{0.1}Ru_{0.9}O₂ (*Joule*, 2023, 7, 558-573), py-RuO₂:Zn (*Nat. Commun.*, 2023, 14, 2517) and a/c-RuO₂ (*Angew. Chem. Int. Ed.* 2021, 60, 2-11), the oxidation state of Ru is lower than +4, and all exhibit excellent OER performance. Here, the mechanism for improving OER activity is different and comprehensive, and not only determined by the oxidation state of Ru.

In our article, multi-scale defects from point to surface in ultrafine RuO₂ crystals reduce the oxidation state of Ru and alter the coordination environment of Ru atoms (**Figure 3**), thereby improving activity by reducing the energy barrier of the OER rate determining step (**Figure 5b**). In addition, both the reduced oxidation state of Ru and extended Ru-O bond weaken the Ru-O interaction, and inhibit the oxidation of lattice oxygen and the dissolution of high-valence Ru, thereby enhancing durability of RuO₂.

Q5. Due to the different half-peak widths of the characteristic peaks attributed to oxygen vacancies in MD-RuO₂-BN and C-RuO₂, it is not possible to determine the relative concentration of oxygen vacancies in the two catalysts from the height of the peaks. The authors should give the ratio of the different characteristic peaks to the whole peak to determine the concentration of oxygen vacancies (Figure 3f). In addition, since the valence state of Ru species in MD-RuO₂-BN has changed, the binding energy position of the Ru-O characteristic peak here should also change.

Reply: Thanks for your valuable comments, this question you raised is very professional. Indeed, using the ratio of the characteristic peak to the whole peak can better reflect the concentration of V_O than directly comparing the intensity of the peaks. Therefore, as suggested, we have provided the ratio of V_O to the whole O 1s peak. As shown in **Figure R30**, the ratio of V_O in RuO₂ and MD-RuO₂-BN is 29.7% and 46.2%, respectively, indicating a significant increase in V_O concentration in the prepared MD-RuO₂-BN. Besides, the binding energy position of the Ru-O characteristic peak here also shifts by about 0.1 eV, further suggesting a redistribution of charges.

In addition, we have replaced **Figure 3f** in the original manuscript with **Figure R29** and made a corresponding revision to the description of **Figure 3f** in the revised manuscript:

“In O 1s spectra (**Figure 3f**), the peak proportion attributable to V_O of MD-RuO₂-BN, C-RuO₂ and K-RuO₂ is 46.2%, 29.7%, and 31.8%, respectively, indicating an obvious increase of V_O concentration in MD-RuO₂-BN. Besides, the binding energy position of the Ru-O characteristic peak for MD-RuO₂-BN also shifts by about 0.1 eV relative to C-RuO₂ and K-RuO₂, further suggesting a redistribution of charges.”

Figure R30. XPS spectra of O 1s for MD-RuO₂-BN and C-RuO₂.

Q6. Both MD-RuO₂-BN and C-RuO₂ have signals attributed to oxygen vacancies in their EPR spectra, and the stronger signal of MD-RuO₂-BN indicated that it contains more oxygen vacancies. The author's statement "offers direct evidence for the existence of VO" should be replaced with the statement that "MD-RuO₂-BN contained more oxygen vacancies than C-RuO₂".

Reply: Thanks a lot for your professional comments. As suggested, we have made a corresponding revision to the description of EPR (**Figure 3g**) in the revised manuscript: “Moreover, MD-RuO₂-BN shows a stronger electron paramagnetic resonance (EPR) signal at $g = 2.003$, also proving that MD-RuO₂-BN contains more V_O than C-RuO₂ and K-RuO₂ (**Figure 3g**).”

Q7. Alterations in the Ru valence state of MD-RuO₂-BN and C-RuO₂ needed to be differentiated on the shift of the absorption edge rather than the white line peak intensity (Figure 3h) (Nat. Commun. 14, 2517 (2023); Nat. Catal. 2, 304-313 (2019)).

Reply: Thanks for your constructive comments. Based on your suggestion, we have marked the shift of the absorption edge in **Figure 3h** (**Figure R31**) and made modifications to the relevant description in the original manuscript:

“The X-ray absorption near edge structure (XANES) of Ru *K*-edge for Ru powder, C-RuO₂ and MD-RuO₂-BN exhibits that the formation of Ru-O bond significantly pushes up the transition energy of XANES, and the absorption edge position for MD-RuO₂-BN is at lower energy compared with that of C-RuO₂ (**Figure 3h**). These indicate that the average valence state of Ru in MD-RuO₂-BN is less than +4, also consistent with XPS analysis results.”

Figure R31. Ru *K*-edge XANES of Ru powder, MD-RuO₂-BN and C-RuO₂.

Q8. What is the reason for the increased hydrophilicity of the MD-RuO₂-BN surface?

Reply: Thanks a lot for your valuable comments. First, the crystal size of MD-RuO₂-BN is much smaller than that of C-RuO₂. The smaller the particle size is, the greater the surface energy. Therefore, the high surface energy of MD-RuO₂-BN determines its stronger hydrophilicity (*Research*, 2019, 1391804; *Angew. Chem.* 2019, 131, 17189-17196; *J. Mater. Chem. A*, 2018, 6, 18384-18388). Moreover, Various defects in catalysts, including vacancies and twinning, may also increase the surface energy of the catalyst. Here, we further calculated the surface energy of RuO₂, RuO₂-V_O, RuO₂-V_{Ru}, RuO₂-T and RuO₂-T-V_{Ru, O} (**Figure R32**). With introducing vacancies and twin grain, MD-RuO₂-BN indeed exhibits higher surface energy, hence showing more hydrophilicity and more accessible sites for H₂O to accelerate subsequent OER.

In addition, we have made a corresponding revision in the manuscript, emphasizing the reason for the increased hydrophilicity of the MD-RuO₂-BN surface. And the relevant figures have been added in the supporting information.

Figure R32. The surface energy of RuO₂, RuO₂-V_O, RuO₂-V_{Ru}, RuO₂-T and RuO₂-T-V_{Ru, O}.

Q9. In the theoretical calculations, what exactly are the sites that enhance the performance of OER, is it the grain boundaries or the reduced particle size, or the presence of vacancies that play a dominant role?

Reply: We thank the reviewer's questions about the theoretical calculations. Various factors such as grain boundaries (*Appl. Surf. Sci.* 2022, 278, 151900), reduced particle size (*Electrochem. Commun.* 2006, 8(9), 1417-1422), or the presence of Ru and O vacancies (*Angew. Chem.* 2021, 34(133), 18969-18977; *Adv. Energy Mater.* 2023, 13(22), 2300615) can influence the catalytic activity. Specifically, in our MD-RuO₂-BN, reduced particle size and the nanoreactor structure together provide highly accessible catalytic sites for OER. These structural features enable MD-RuO₂-BN to exhibit enhanced efficiency in catalyzing OER.

Moreover, we have further optimized the electronic structure by incorporating Ru, O vacancies and twin grain boundaries. Our calculations indicate that these modifications favorably impact the OER performance in the context of the nanoreactor structures. However, it is challenging to determine the individual contribution of each factor to the overall enhancement in OER activity. Our findings suggest that the synergistic effect of these combined factors significantly improves the OER performance. Consequently, attributing the observed enhancements to a single factor is difficult. With the improvement of characterization technology, this work will be carried out and solved in the future. We believe that it is of importance for the electronic tailoring under a good structural design.

Q10. There are a large number of grammatical errors and data results analysis errors in the manuscript. Authors need to check the manuscript carefully.

Reply: Thank you for reviewing our manuscript so carefully. We think it is very helpful to improve the quality of our paper. As suggested, we have carefully checked the manuscript and made modifications to the relevant grammatical errors, and marked them in the revised manuscript. In addition, we have further confirmed the correctness and rationality of all data and their analysis results to avoid errors similar to those made

in questions 3 and 4.

Finally, thank you again for your efforts in reviewing our article, which really benefited us a lot. We sincerely hope that this revision can receive your recognition and would be happy to hear any comments and opinions you may have on the current version of this article.

REVIEWER COMMENTS

Reviewer #1 (Remarks to the Author):

First, I would like to complement the authors on significantly improving the manuscript. The additional experiments, particularly the completely reworked electrolyzer section, have made this a well-executed piece of work. The extra experiment in KCl that yielded larger nanoparticles is also extremely valuable. The results are intriguing and seem to defy previous research.

In particular, I am perplexed by the better electrochemical stability of the small nanoparticles (LiCl-KCl flux) compared to the larger nanoparticles (KCl). There is a substantial body of evidence suggesting that the lower the oxidation state of Ru, the more prone it is to being unstable and gradually corroding/dissolving. See for example somewhat older but perfectly executed and valid papers [DOI: 10.1016/j.cattod.2015.08.014 and DOI: 10.1002/celec.201402262]).

Therefore, I do not understand why the ICP results show the larger particles have more of a tendency toward dissolution than the smaller nanoparticles. Dissolution is a chemical issue that cannot be rationalized by the DFT calculations the authors carried out. In fact, from my interpretation of the DFT, the authors seem to believe grain boundary formation somehow improves stability, which appears to contradict literature showing fully oxidized RuO₂ tends to be most stable. Furthermore, the authors' interpretation of the XPS and XAS results, and their statement "Both the lowered Ru valence state and weakened Ru-O interaction would inhibit the oxidation of lattice oxygen and the dissolution of high-valence Ru, resulting in greatly enhanced durability," seems to contradict previous rigorous experimental work suggesting exactly the opposite. My recommendation is to take samples for ICP studies at regular intervals (every day) while running extended chronoamperometry experiments for at least 7 days.

Reviewer #2 (Remarks to the Author):

The author's revisions explain my questions well, thus it is recommended to accept this paper to publish on Nature Communications.

Response to Reviewer #1

First, I would like to complement the authors on significantly improving the manuscript. The additional experiments, particularly the completely reworked electrolyzer section, have made this a well-executed piece of work. The extra experiment in KCl that yielded larger nanoparticles is also extremely valuable. The results are intriguing and seem to defy previous research.

In particular, I am perplexed by the better electrochemical stability of the small nanoparticles compared to the larger nanoparticles. There is a substantial body of evidence suggesting that the lower the oxidation state of Ru, the more prone it is to being unstable and gradually corroding/dissolving. See for example somewhat older but perfectly executed and valid papers (DOI: 10.1016/j.cattod.2015.08.014; DOI: 10.1002/celec.201402262).

Therefore, I do not understand why the ICP results show the larger particles have more of a tendency toward dissolution than the smaller nanoparticles. Dissolution is a chemical issue that cannot be rationalized by the DFT calculations the authors carried out. In fact, from my interpretation of the DFT, the authors seem to believe grain boundary formation somehow improves stability, which appears to contradict literature showing fully oxidized RuO₂ tends to be most stable. Furthermore, the authors' interpretation of the XPS and XAS results, and their statement "Both the lowered Ru valence state and weakened Ru-O interaction would inhibit the oxidation of lattice oxygen and the dissolution of high-valence Ru, resulting in greatly enhanced durability," seems to contradict previous rigorous experimental work suggesting exactly the opposite. My recommendation is to take samples for ICP studies at regular intervals (every day) while running extended chronoamperometry experiments for at least 7 days.

Reply: We really appreciate your recognition and advices on our revised manuscript. We have re-demonstrated the ICP test and further revised our manuscript by addressing the remaining issues you raised. Through resolving these issues, we hope that the revised version can receive your favorable consideration.

Your current confusion is that small nanoparticles (LiCl-KCl flux) have better electrochemical stability than large nanoparticles (KCl). For this, we believe that particle size is not the main factor determining the stability of RuO₂ catalysts. Here, the formation of grain boundaries and vacancies effectively regulates the electronic structure and coordination environment of Ru. Therefore, both the lowered Ru valence state and weakened Ru-O interaction would inhibit the oxidation of lattice oxygen and the dissolution of high-valence Ru, resulting in greatly enhanced durability. We noticed that you believe this conclusion is contrary to previous experimental work (DOI: 10.1016/j.cattod.2015.08.014; DOI: 10.1002/celec.201402262). After carefully reading the two articles you recommended, we believe that these articles only clearly indicate that the stability of RuO₂ is better than that of the metal Ru, which is currently recognized. However, there was an absence in discussion on which Ru with the high oxidation state or low oxidation state is more stable in RuO_x. To our knowledge, in many reported articles, RuO₂ with lower oxidation states also exhibits excellent OER stability and activity (*Nat. Commun.*, 2022, 13, 3784; *Angew. Chem. Int. Ed.* 2021, 60, 2-11; *Joule*, 2023, 7, 558-573; *Nat. Commun.*, 2023, 14, 2517). They also indicate that the decrease in the valence state of Ru (< +4) can inhibit the dissolution of high valent Ru and the oxidation of lattice oxygen during the OER process. These all support our conclusion.

Furthermore, we have re-demonstrated the ICP test based on your constructive advices. Specifically, the chronopotentiometric measurements of MD-RuO₂-BN and K-RuO₂ (mass loading of 0.1 mg on glassy carbon electrode) for 7 days at a constant current density of 10 mA cm⁻² was conducted in a conventional three-electrode system (**Figure R1**). Based on regular monitoring (the time interval was 24 h) the Ru concentration in 0.5 M H₂SO₄ electrolyte (25 mL) by ICP-OES (Agilent ICPOES700, **Figure R2**), the dissolution of Ru was gradually collected (**Figure R3**). For testing steps, we first prepared the standardization curve for the Ru standard solution (take 0, 0.25, 0.5, 1.0, 2.5 mL of 100 mg/L Ru standard solution, respectively, into a 50 mL volumetric bottle, then add 5% dilute nitric acid to steady volume to 50 mL and shake

well), the obtained standardization curve is shown in the **Figure R4**. Then we tested the collected electrolyte in sequence, and finally determine the final content of the measured Ru element in each sample through the spectrogram (the relevant raw data is provided in **Table R1**). After calculation, as shown in **Table R2**, during continuous OER process, the dissolution rate of Ru in K-RuO₂ increased from 4.93 % to 7.89 %, while it is only increased from 3.29 % to 5.59 % for MD-RuO₂-BN. Obviously, the dissolution of Ru in MD-RuO₂-BN is significantly inhibited compared to K-RuO₂, which is consistent with previous ICP test results, further confirming our conclusion above.

In addition, we have made a corresponding revision in the manuscript, and the relevant figures (**Figure S30-31** with supplementary note) and table (**Table S2**) have been added in supporting information.

The revision is shown as below:

(Page 16) “ICP-OES studies at regular intervals (24 h) while running extended chronopotentiometry experiments for 7 days (Figure S30-31, specific experimental details can refer to Supporting Information) further indicate that the dissolution of Ru in MD-RuO₂-BN is significantly inhibited compared to K-RuO₂ (Table S2).”

Figure R1. Photographs of the conventional three-electrode system.

Figure R2. Photographs of the ICP-OES (700 Series, Agilent Technologies).

Figure R3. Photographs of electrolytes collected by step-by-step (1-1~1-7 for MD-RuO₂-BN; 2-1~2-7 for K-RuO₂), the time interval was 1 day (24 h), a total of 7 days.

Figure R4. The obtained standardization curve of Ru standard solution.

Table R1. Raw data related to ICP-OES testing.

Sample number	Measured element	Dilution factor	Instrument reading	Final content	Unit
1-1	Ru	5	0.0197	0.10	mg/L
1-2	Ru	5	0.0215	0.11	mg/L
1-3	Ru	5	0.0233	0.12	mg/L
1-4	Ru	5	0.0294	0.15	mg/L
1-5	Ru	5	0.0303	0.15	mg/L
1-6	Ru	5	0.0319	0.16	mg/L
1-7	Ru	5	0.0332	0.17	mg/L
2-1	Ru	5	0.0301	0.15	mg/L
2-2	Ru	5	0.0340	0.17	mg/L
2-3	Ru	5	0.0381	0.19	mg/L
2-4	Ru	5	0.0387	0.19	mg/L
2-5	Ru	5	0.0401	0.20	mg/L
2-6	Ru	5	0.0443	0.22	mg/L
2-7	Ru	5	0.0481	0.24	mg/L

Table R2. ICP investigation at regular intervals (24 h) during OER processes catalyzed by MD-RuO₂-BN and K-RuO₂, respectively.

Samples	Measured element	Content (mg/L)	Dissolution rate
MD-RuO ₂ -BN-1	Ru	0.10	3.29 %
MD-RuO ₂ -BN-2	Ru	0.11	3.62 %
MD-RuO ₂ -BN-3	Ru	0.12	3.95 %
MD-RuO ₂ -BN-4	Ru	0.15	4.93 %
MD-RuO ₂ -BN-5	Ru	0.15	4.93 %
MD-RuO ₂ -BN-6	Ru	0.16	5.26 %
MD-RuO ₂ -BN-7	Ru	0.17	5.59 %
K-RuO ₂ -1	Ru	0.15	4.93 %
K-RuO ₂ -2	Ru	0.17	5.59 %
K-RuO ₂ -3	Ru	0.19	6.25 %
K-RuO ₂ -4	Ru	0.19	6.25 %
K-RuO ₂ -5	Ru	0.20	6.58 %
K-RuO ₂ -6	Ru	0.22	7.24 %
K-RuO ₂ -7	Ru	0.24	7.89 %

Supplementary Note: The chronopotentiometric measurements were performed in a conventional three-electrode system at room temperature. The OER current density was set at 10 mA cm⁻². The catalyst loading was all 0.1 mg. The electrolyte was 25 mL of 0.5 M H₂SO₄. The time interval for collection was 24 h.

REVIEWERS' COMMENTS

Reviewer #1 (Remarks to the Author):

The authors addressed my concerns and the article is acceptable for the publication.